# Ultrastructural changes in cardiac and skeletal myoblasts following *in vitro* exposure to monensin, salinomycin, and lasalocid

**Danielle Henn**[1☉], **Antonia V. Lensink**[2☉]*, **Christo J. Botha**[1]

**1** Department of Paraclinical Sciences, Faculty of Veterinary Science, University of Pretoria, Pretoria, South Africa, **2** Electron Microscope Unit, Department of Anatomy and Physiology, Faculty of Veterinary Science, University of Pretoria, Pretoria, South Africa

☉ These authors contributed equally to this work.
* antoinette.lensink@up.ac.za

**Data Availability Statement:** All files are available from the Institutional Repository of the University

## Abstract

Carboxylic ionophores are polyether antibiotics used in production animals as feed additives, with a wide range of benefits. However, ionophore toxicosis often occurs as a result of food mixing errors or extra-label use and primarily targets the cardiac and skeletal muscles of livestock. The ultrastructural changes induced by 48 hours of exposure to 0.1 μM monensin, salinomycin, and lasalocid in cardiac (H9c2) and skeletal (L6) myoblasts *in vitro* were investigated using transmission electron microscopy and scanning electron microscopy. Ionophore exposure resulted in condensed mitochondria, dilated Golgi apparatus, and cytoplasmic vacuolization which appeared as indentations on the myoblast surface. Ultrastructurally, it appears that both apoptotic and necrotic myoblasts were present after exposure to the ionophores. Apoptotic myoblasts contained condensed chromatin and apoptotic bodies budding from their surface. Necrotic myoblasts had disrupted plasma membranes and damaged cytoplasmic organelles. Of the three ionophores, monensin induced the most alterations in myoblasts of both cell lines.

## Introduction

Carboxylic ionophores are polyether antibiotics extensively used in production animals for the promotion of growth and feed efficiency, as well as for the control of coccidiosis [1–4]. However, concomitant with the widespread use of ionophores, ionophore toxicity has also been reported. The cardiac and skeletal muscles are the primary targets affected, with animals showing various clinical signs, including tachycardia, arrhythmias, dyspnea, depression, ataxia, hypoactivity, and weakness [5–10].

Carboxylic ionophores can form zwitterionic complexes with cations, and transport these cations across biological membranes, which may alter the intracellular ion homeostasis and disrupt various cellular processes [3, 11–14]. Toxic concentrations of ionophores lead to increased intracellular calcium [3, 11–13, 15–18], disruption of the mitochondrial membrane potential [11–13, 17, 19, 20] as well as oxidative phosphorylation [12, 13, 17], production of

of Pretoria database (URL: https://repository.up.ac.za/).

**Funding:** Health and Welfare Sector Education and Training Association (HWSETA), N02015: B_BOTH_HENN The funders had no role in study design, data collection and analysis, decision to publish, or preparation of the manuscript.

**Competing interests:** The authors have declared that no competing interests exist.

reactive oxygen species [17–21], inhibition of cellular protein transport [14–16, 19], alteration of intracellular pH [3, 12, 16] and increased lipid peroxidation [3].

Lesions associated with ionophore toxicosis vary from species to species and can take time to develop. Cardiac and skeletal muscle fibers of animals suffering from ionophore toxicosis undergo a process of degeneration, necrosis, and attempts at repair, with a variable inflammatory component [6, 22–24]. In affected myocytes of swine fed a lethal dose of monensin, disrupted contractile material, swollen mitochondria, and sarcoplasmic vacuolization were observed [23, 24]. Hepatocytes undergo similar ultrastructural changes, with an increased surface of smooth endoplasmic reticulum and several lipid droplets [25].

Moreover, in vitro ionophore exposure induces mitochondrial condensation, dilation of the Golgi apparatus, and excessive vesiculation of the cytoplasm of cells and tissues [14, 16, 26]. Ionophores promote both apoptosis and programmed necrosis through disruption of the mitochondrial membrane potential [18–21, 27, 28]. Autophagy plays a protective role during ionophore exposure, and its inhibition results in increased cell death [19–21]. The main (programmed / apoptotic and non-programmed / necrotic) cell death mechanisms have distinct morphological phenotypes and can be differentiated with imaging modalities such as electron microscopy. Apoptosis is characterized by cell rounding and shrinkage, plasma membrane smoothing, apoptotic body (vesicles of a relatively large size, 1 to 5μm, with a range of composition and structure) formation, chromatin condensation (pyknosis), nuclear fragmentation (karyorrhexis), and mitochondrial changes amongst others [19–20, 29–31]. Primary and secondary (programmed) necrosis result in similar morphological features including cellular and cytoplasmic organelle swelling, vacuolization, moderate chromatin condensation, and plasma membrane rupture with the subsequent release of the intracellular contents [30, 31]. In this study, ultrastructural changes occurring in cardiac and skeletal myoblasts following exposure to 0.1 μM monensin, salinomycin, and lasalocid were investigated using electron microscopy.

## Materials and methods

### Cell culture and exposure

Rat cardiac (H9c2(2–1) (ATCC® CRL-1446™)) and skeletal (L6 (JCRB9081)) muscle cell lines were obtained from the American Type Culture Collection and the Japanese Collection of Research Bioresources Cell Bank, respectively. Myoblasts were cultured in Dulbecco's Modified Eagle's medium (DMEM) (PAN Biotech) supplemented with 10% fetal bovine serum (FBS) (Gibco), 100 U penicillin/ml, and 100 U streptomycin/ml (Lonza), and then incubated at 37˚C in a humidified atmosphere with 5% $CO_2$ [32].

Monensin sodium (MW: 692.85 g/mol), salinomycin sodium (MW: 772.98 g/mol) and lasalocid A sodium (MW: 612.77 g/mol) were obtained from Dr Ehrenstorfer™. Stock solutions were prepared by dissolving the ionophores in methanol (MeOH) up to a concentration of 40 mM [33].

The cytotoxicity of the ionophores was determined using a modified MTT viability assay previously described [33]. Briefly; myoblasts were exposed to a serial dilution of the ionophores for 24, 48, and 72 h. Post-exposure, the plates were washed, followed by the addition of 200 μl complete media and 20 μl (5 mg/ml in PBS) MTT (Sigma-Aldrich) per well and incubated for 2 h at 37˚C. Following incubation, the medium was removed, 100 μl dimethyl sulfoxide (DMSO) was added and the absorbance was measured. The background absorbance was subtracted and the viability of the myoblasts was expressed as a percentage of the solvent control.

The cell lines were seeded at a concentration of 100 000 myoblasts/ml, either into a 6-well plate (for transmission electron microscopy (TEM)) or onto sterile 10 mm coverslips in a

24-well plate (for scanning electron microscopy (SEM)), and allowed 24 h to adhere and stabilize. The myoblasts were exposed to 0.1 μM monensin, salinomycin, and lasalocid for 48 h. This concentration was selected based on previous cytotoxicity experiments as this concentration falls within the intermediate range between the $EC_{50}$s of the three ionophores (S1–S4 Figs) [33].

## Sample preparation for transmission and scanning electron microscopy

After exposure, myoblasts were fixed with 2.5% glutaraldehyde in 0.075 M phosphate buffer (pH 7.4) for at least 1 h. The fixed myoblasts were centrifuged in microtubes and the pellets were rinsed three times with 0.075 M phosphate buffer for 10 min each. A second fixation step involving a 1% osmium tetroxide ($OsO_4$) solution for 1 h, was performed [34]. This was followed by washing the pellets thrice with distilled water. The samples were dehydrated with increasing ethanol (EtOH) concentrations (50, 70, 90, 96, and 100% EtOH) for 10 min each, and finally with 100% EtOH for 1 h. EtOH was replaced with propylene oxide for 10 min and slowly infiltrated with an epoxy-resin mixture. The samples were first incubated with 2:1 propylene oxide/epoxy resin, followed by 1:2 propylene oxide/epoxy resin, for 1 h each. The resin mixture was replaced with pure epoxy for 2–3 h, then embedded in TAAB 812 epoxy resin [35] and left in an oven at 65˚C overnight to polymerize. The resin blocks were cut into ultrathin sections (~ 100 nm thick) using a Leica EM UC7 microtome and placed on a $300 \times 75$ mesh copper grid (Agar Scientific). Each sample was stained for 6 min with uranyl acetate and for 3 min with lead citrate [36]. Samples were viewed using a JEOL JEM 1400-FLASH transmission electron microscope (Tokyo, Japan).

For SEM, coverslips were fixed and dehydrated as described above. The coverslips were then incubated with a 1:1 mixture of EtOH and 1,1,1,3,3,3-hexamethyldisilazane (HMDS) (Merck), followed by HMDS for 30 min at room temperature. Finally, coverslips were left overnight in 100% HMDS. The coverslips were attached to an aluminum stub using 12 mm carbon adhesive tabs (Electron Microscopy Sciences) and coated with chromium using a Quorum Q150T ES sputter coater (East Sussex, United Kingdom). The samples were viewed using a Zeiss SUPRA 55VP scanning electron microscope (Oberkochen, Germany).

## Ethics statement

This project was approved by the Research Ethics Committee of the Faculty of Veterinary Science, University of Pretoria (REC070-19, 2019/05/06).

## Results

### H9c2 myoblasts

H9c2 myoblasts are large tapered cells with oval-shaped nucleinear the center of the myoblast. The cytoplasm contained small vesicles, rough endoplasmic reticulum (RER), and the Golgi apparatus located in proximity to the nucleus ( Fig 1A–1D). SEM analysis of the surface of H9c2 myoblasts revealed that when attached to a coverslip, the myoblasts were generally large, flattened, and mat-like (Fig 1E–1G). The nuclei can be seen lying beneath the surface of the myoblasts, in addition to smaller bulges, believed to be cellular organelles, distributed throughout the cytoplasm. The surface of spread myoblasts was relatively smooth with only a few small finger-like projections (Fig 1E–1G). In contrast, myoblasts with a more three-dimensional shape were covered with filipodia- and bleb-like surface structures (Fig 1H, see inset). A few cracks due to a drying artifact were observed on the surface of the myoblasts.

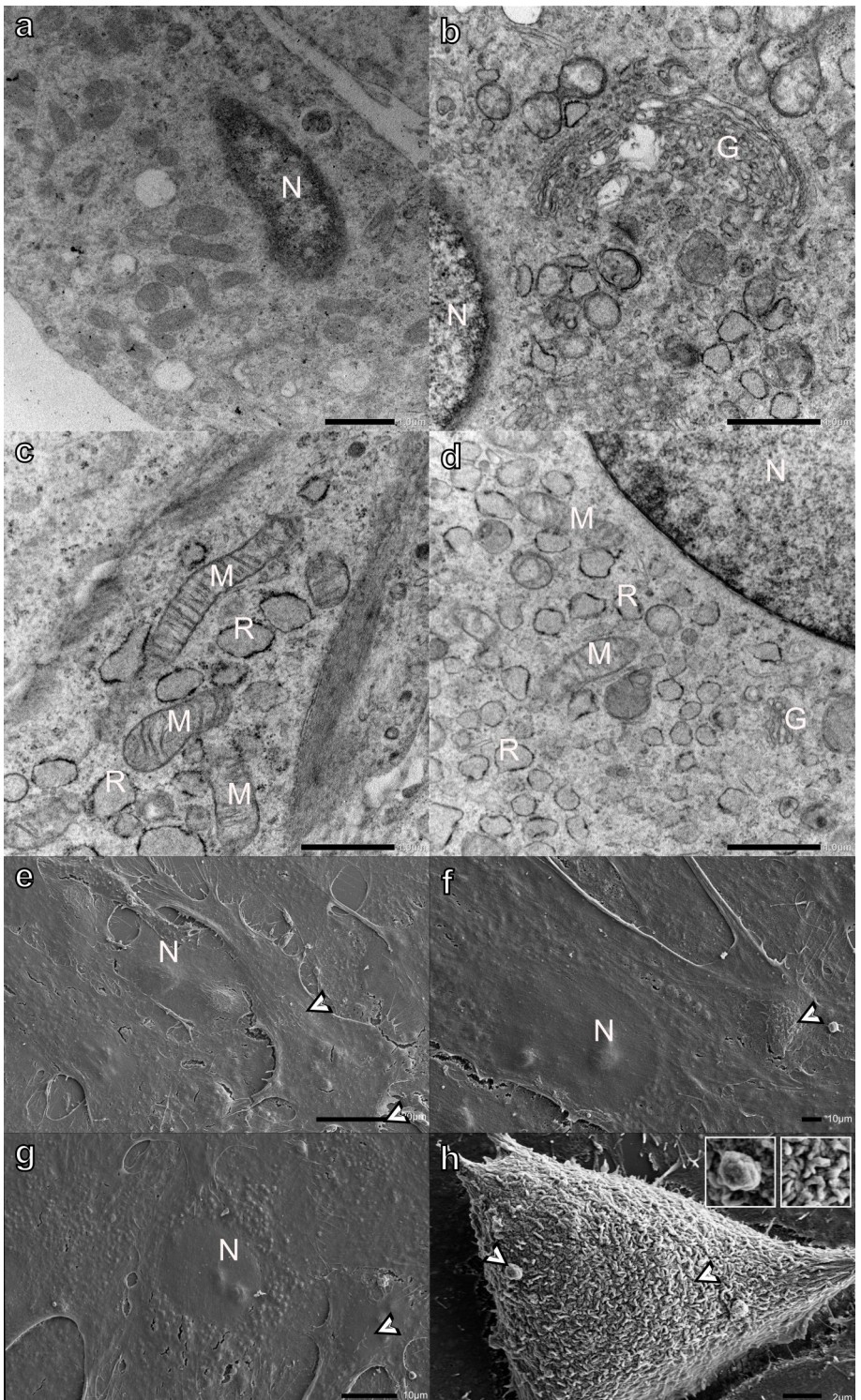

**Fig 1.** Transmission (a-d) and scanning (e-h) electron micrographs of H9c2 myoblasts incubated in DMEM for 48 h (negative control). Bordered arrowheads- filipodia- and bleb-like surface structures (see inset), G-Golgi apparatus, M-mitochondria, N-nucleus, and R-rough endoplasmic reticulum. Scale bars: a-d = 1 μm, e = 20 μm, f-g - = 10 μm, h = 2 μm.

Monensin had the greatest effect on myoblasts, with the majority of cells being filled with large electron-lucent vacuoles (Fig 2A–2D). Extensive mitochondrial morphological changes were also observed. Several myoblasts contained condensed mitochondria as well as mitophagy-like structures enclosed in vacuoles (Fig 2B). The presence of membranes concentrically organized in onion-like patterns, characteristic inner and outer mitochondrial membrane arrangements, and in some instances mitochondria with recognizable, albeit, degraded cristae (Fig 2B and 2B inset), could be indicative of mitochondria in variable stages of vacuolar degeneration (Fig 2B), which may suggest that the mitochondria are contributing to the extensive cytoplasmic vacuolization observed. The RER was distended (Fig 2B and 2C) and some myonuclei presented condensed and marginalized chromatin (Fig 2C). The myoblasts contained variable amounts of autophagic vesicles recognizable by their multivesicular myelin-like, or granular osmiophilic content (Fig 2C, inset). Necrotic myoblasts with a loss of membrane integrity and degradation of the cytoplasmic components were also observed (Fig 2D). Monensin exposure resulted in large indentations on the surface of myoblasts, giving them a pockmarked appearance (Fig 2E). These 'pockmarks' were visible over the entire myoblast surface except at the nuclear position. The indentations are thought to be a dehydration and drying artifact resulting from the collapse of the large number of vacuoles present in these cells (Fig 2A–2D). A number of myoblasts were 'rounded-off,' with a variable loss of surface structures (Fig 2F), while others were in more advanced stages of apoptosis (Fig 2G). Advanced apoptosis could be seen by the extensive formation and blebbing of apoptotic bodies (vesicles of a relatively large size, between 1 to 5 μm, with a wide range of composition and structure) (Fig 2G). Furthermore, several necrotic myoblasts, primarily consisting of cellular debris and exhibiting a nearly complete loss of membrane continuity, were observed (Fig 2H).

Salinomycin affected myoblasts to a lesser degree when compared with the monensin-exposed cells. The Golgi apparatus was dilated; however, fewer myoblasts showed excessive vacuolization (Fig 3A–3C). Several autophagic vesicles were also observed (Fig 3D). The mitochondria and RER remained largely unaffected (Fig 3D). A few myoblasts also exhibited surface indentations giving the cell a pockmarked appearance similar to H9c2 myoblasts following monensin exposure (Fig 3E and 3F). Individual apoptotic (Fig 3G) and necrotic (not shown) myoblasts were also observed. However, the surfaces of the majority of myoblasts remained unaffected with filipodia- and bleb-like surface structures (Fig 3H).

Lasalocid had the least effect on myoblast ultrastructure (Fig 4A and 4B). A few myoblasts had swollen Golgi apparatus and a few electron-lucent vacuoles and autophagic vesicles. The Golgi apparatus generally remained flat at the cis-face, whereas the cisternae at the trans-face became swollen (Fig 4C and 4D). SEM analysis confirmed this finding, with the majority of myoblasts appearing unaffected, with typical sizes, shapes, and presenting with filipodia- and bleb-like surface structures after lasalocid exposure (Fig 4E–4G). Occasionally a few myoblasts with shallow indentations near the nucleus were found (Fig 4H).

In summary, after ionophore exposure, the Golgi apparatus became dilated and the cytoplasm was filled with electron-lucent vesicles. Additionally, the mitochondria of the more severely affected myoblasts were condensed. The RER was mostly unaffected, with a homogenous content, and the ribosomes remained attached to the membrane.

## L6 myoblasts

L6 myoblasts are round-to-spindle-shaped cells and are smaller than the H9c2 myoblasts. Myonuclei were typically centrally positioned, often with one or more adjacent Golgi apparatus visible (Fig 5A–5D). The mitochondria and RER are distributed throughout the cytoplasm. Additionally, small filipodia-like protrusions were observed extending from the myoblast

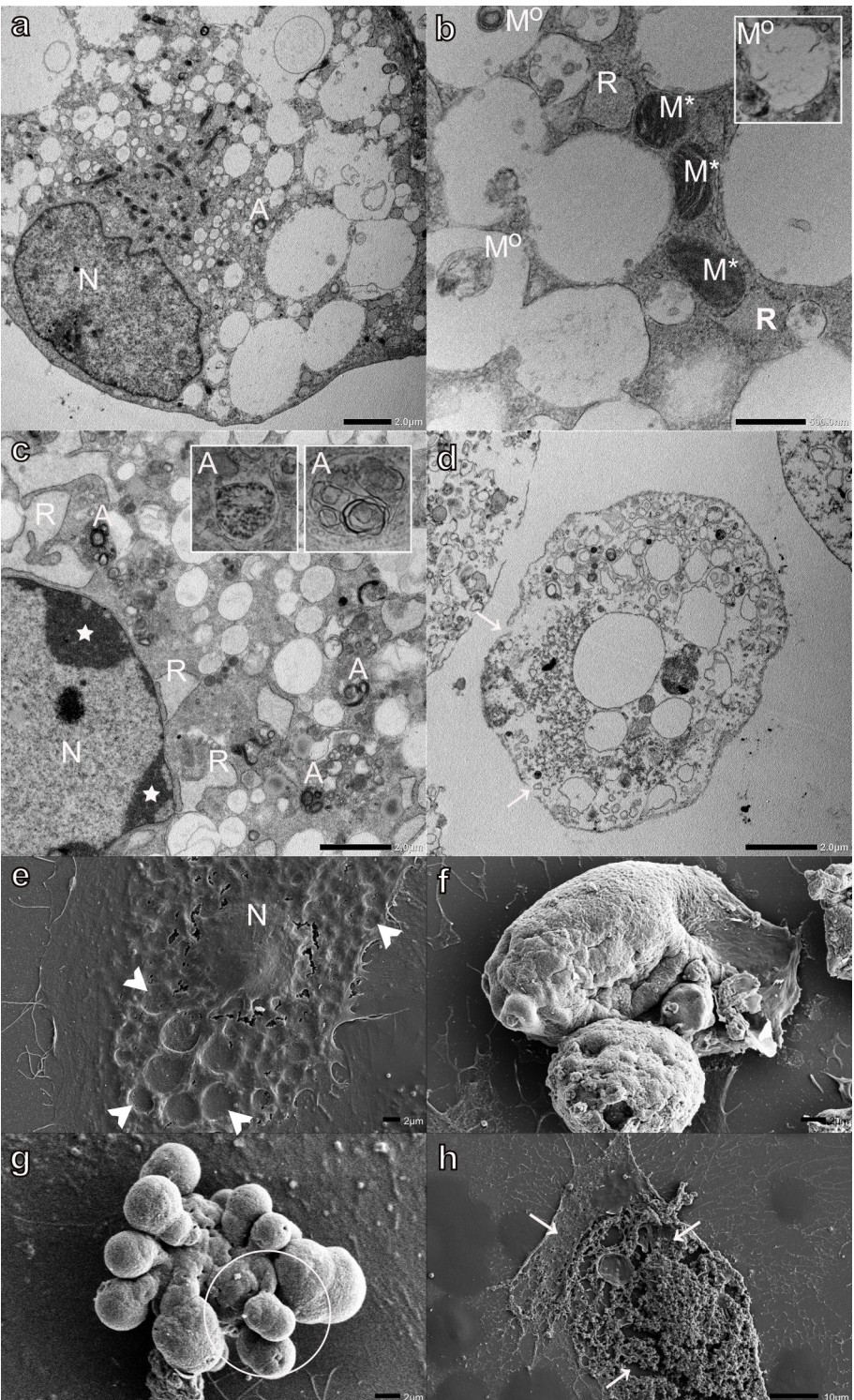

**Fig 2.** Transmission (a-d) and scanning (e-h) electron micrographs of H9c2 myoblasts exposed to 0.1 µM monensin for 48 h. Arrows-loss of membrane integrity, arrowheads-surface indentations, circled-apoptotic bodies, stars-condensed and marginalized chromatin, A-autophagic vesicles, M°- mitochondrial vacuolar degeneration, M*-condensed mitochondria, N-nucleus, and R-rough endoplasmic reticulum. Scale bars: a, c-d = 1 µm, b = 500 nm, e-g = 2 µm, h = 10 µm.

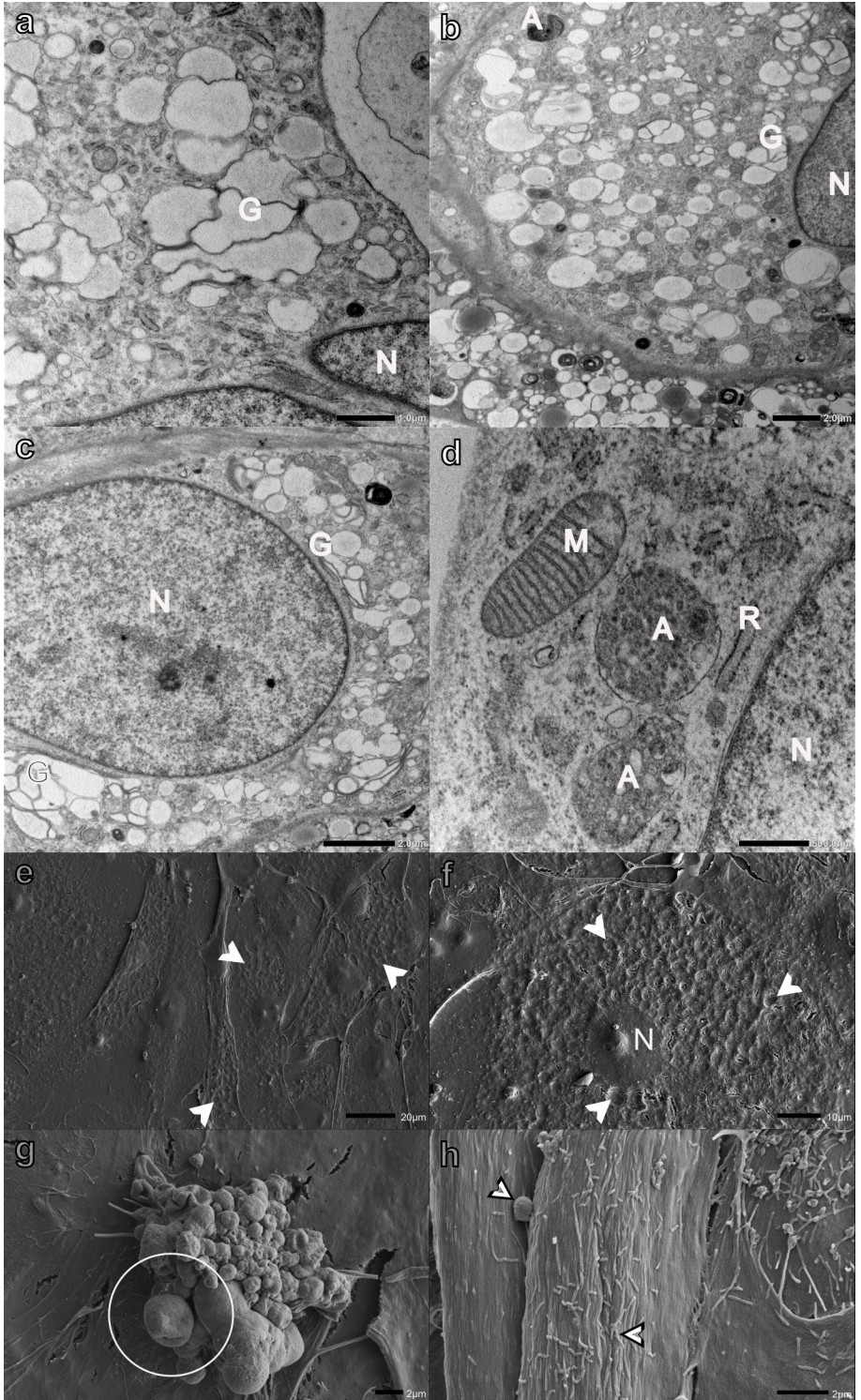

**Fig 3.** Transmission (a-d) and scanning (e-h) electron micrographs of H9c2 myoblasts exposed to 0.1 µM salinomycin for 48 h. Arrowheads-surface indentations, bordered arrowheads- filipodia- and bleb-like surface structures, circled-apoptotic bodies, A-autophagic vesicles, G-Golgi apparatus, M-mitochondria, N-nucleus, and R-rough endoplasmic reticulum. Scale bars: a = 1 µm, b-c, g-h = 2 µm, d = 500 nm, e = 20 µm, f = 10 µm.

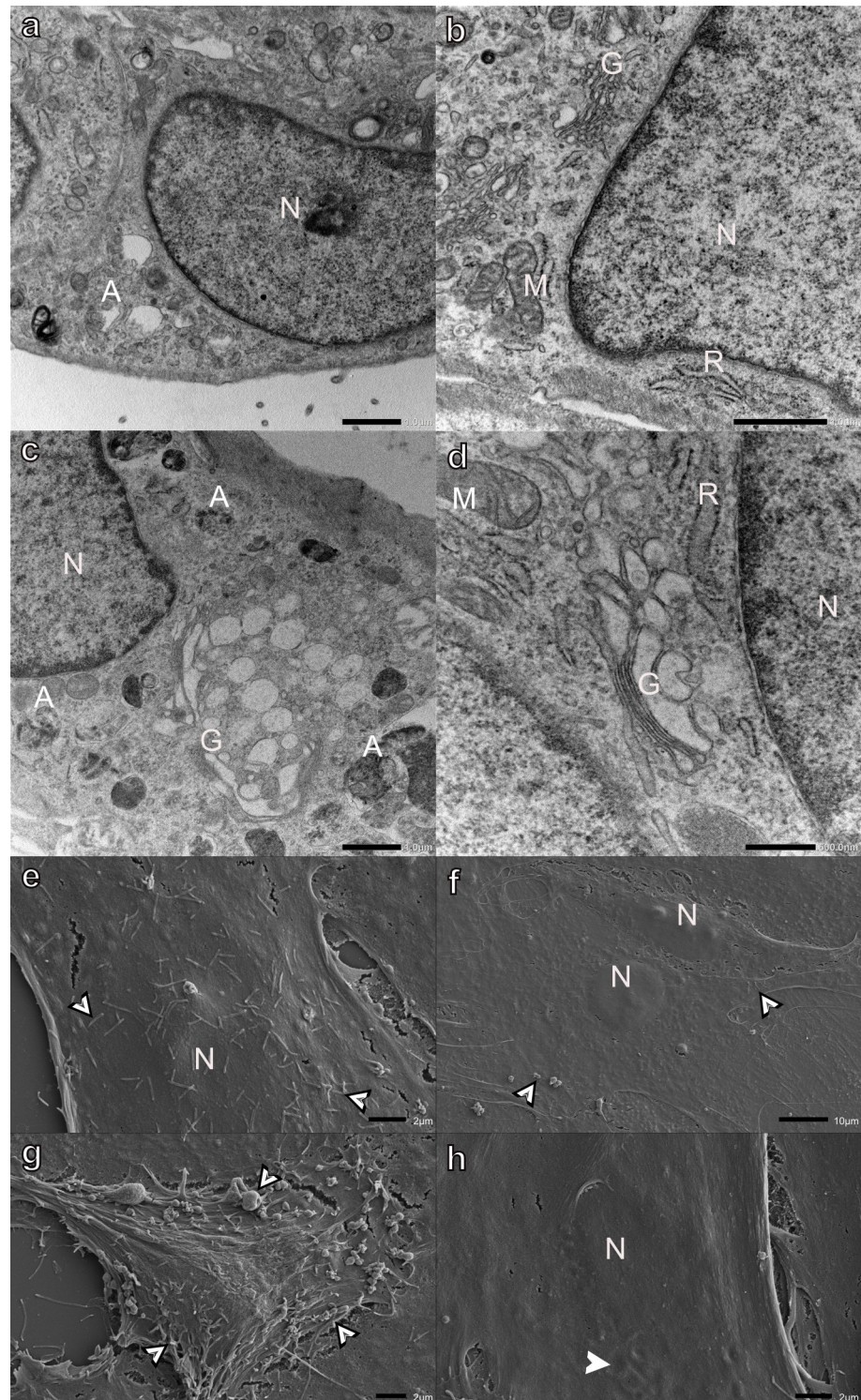

**Fig 4.** Transmission (a-d) and scanning (e-h) electron micrographs of H9c2 myoblasts exposed to 0.1 µM lasalocid for 48 h. Arrowheads-surface indentations, bordered arrowheads-filipodia- and bleb-like surface structures, A-autophagic vesicles, G-Golgi apparatus, M-mitochondria, N-nucleus, and R-rough endoplasmic reticulum. Scale bars: a-c = 1 µm, d = 500 nm, e, g-h = 2 µm, f = 10 µm.

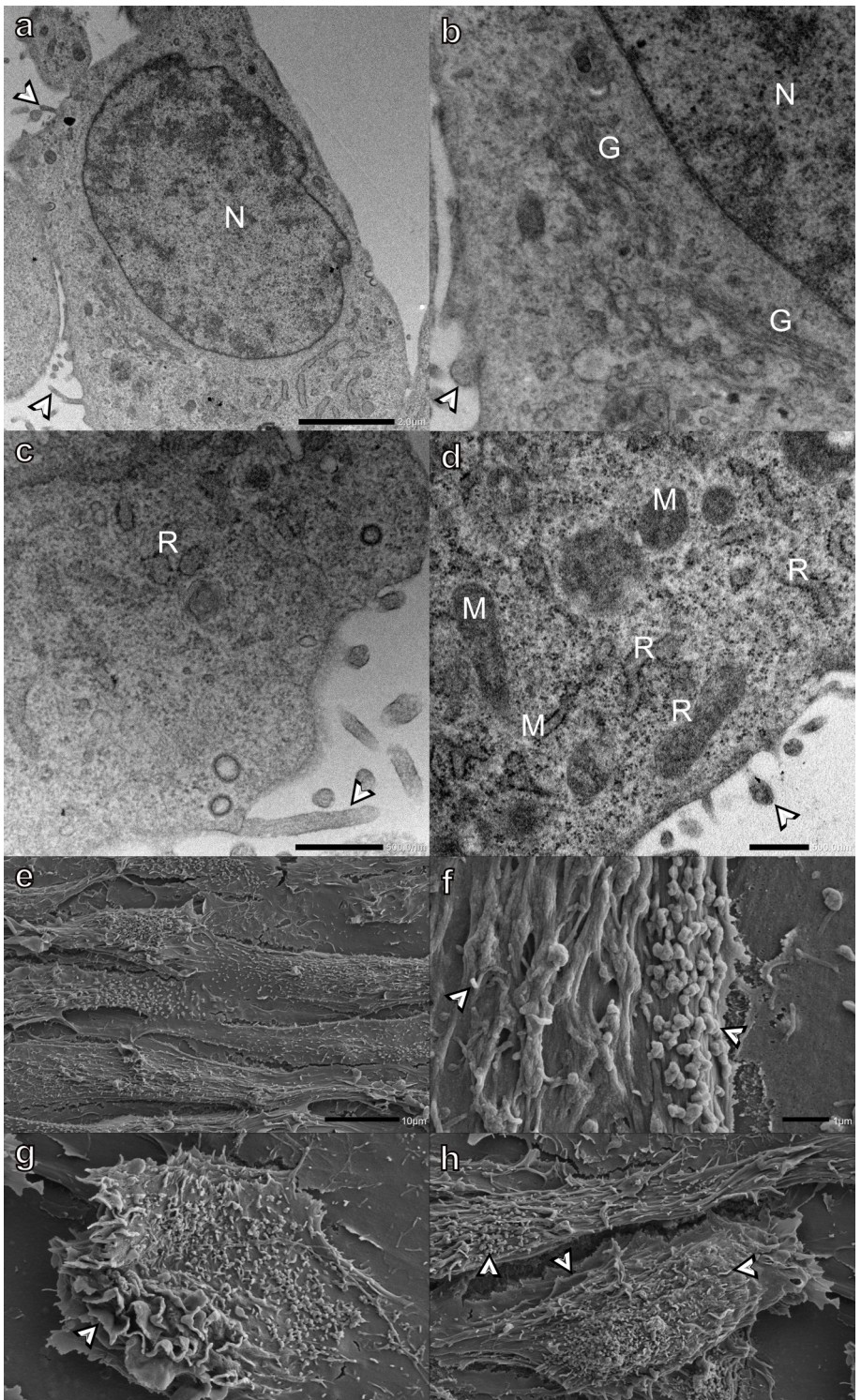

**Fig 5.** Transmission (a-d) and scanning (e-h) electron micrographs of L6 myoblasts incubated in DMEM for 48 h. Bordered arrowheads-filipodia-, bleb-, or frill-like surface structures, A-autophagic vesicles, G-Golgi apparatus, M-mitochondria, N-nucleus, and R-rough endoplasmic reticulum. Scale bars: a, g-h = 2 μm, b-d = 500 nm, e = 10 μm, f = 1 μm.

plasma membrane (Fig 5C). Surface analysis of L6 myoblasts revealed elongated and rounded three-dimensional myoblasts covered with surface structures (Fig 5E–5H). The surface structures were either small, filipodia-like, bulkier bleb-like, or ridges resembling frills. Damage due to drying artifacts was also observed.

After monensin exposure, myoblasts were filled with electron-lucent vacuoles (Fig 6A) and either elongated (Fig 6A) or condensed mitochondria (Fig 6B) were observed. The nuclei appeared slightly lobulated compared to the controls, with varying degrees of condensed chromatin accumulating at the inner nuclear membrane (marginalization) (Fig 6C). Autophagic vesicles (Fig 6D) were present throughout the cytoplasm of the exposed myoblasts. Affected myoblasts lost their surface structures, and had condensed cytoplasm, and a few myoblasts were observed shedding apoptotic bodies (Fig 6D and 6G). Monensin exposure, similar to what was seen in the H9c2 myoblasts, resulted in deep indentations on the surface of many myoblasts, giving the myoblasts a pockmarked appearance (Fig 6E). There was an increase in the number of rounded myoblasts that either exhibited a smooth membrane that lacked surface structures (Fig 6F) or were apoptotic (Fig 6G). A few necrotic myoblasts with extensive loss of cell membrane integrity were observed (Fig 6H).

Myoblasts exposed to salinomycin contained electron-lucent vacuoles, however, less cells were as extensively vacuolated when compared to cells exposed to monensin (Fig 7A). Similarly, the Golgi apparatus showed swelling and dilation, but not as severe as what was observed with monensin exposure (Fig 7B and 7C). Some nuclei contained condensed chromatin and were slightly lobulated (Fig 7D). Mitochondria and RER were largely unaffected. The majority of myoblasts, visualized with SEM, were unaffected after salinomycin exposure (Fig 7E–7G), with filipodia-, bleb-, and frill-like structures present on the cell surface (Fig 7E–7G). A few myoblasts presented with indentations giving the cells a pockmarked appearance (Fig 7F and 7G). Isolated incidences of rounded myoblasts lacking surface structures (Fig 7H), as well as individual apoptotic myoblasts (not shown), were observed.

Lasalocid exposure had the least effect on the L6 ultrastructure, with most myoblasts retaining their normal ultrastructural characteristics. A few myoblasts contained electron-lucent vacuoles, individual condensed mitochondria and a few autophagic vesicles (Fig 8A–8D). One or two myoblasts with apoptotic body formation were also observed (Fig 8D and 8E). The majority of myoblasts resembled those of the control and were elongated or rounded, with many surface structures (Fig 8F and 8G), except a few isolated cells presenting with shallow indentations (Fig 8H).

Concisely, similar to H9c2 myoblasts, the major ultrastructural changes that occurred due to ionophore exposure were vacuolization of the cytoplasm as well as mitochondrial condensation.

## Discussion

After ionophore exposure, pronounced ultrastructural changes were observed in both H9c2 and L6 cell lines. All three ionophores caused similar ultrastructural changes in both myoblast cell lines, though to varying extents and severities. Of the three ionophores, monensin had the greatest effect on myoblast ultrastructure, followed by salinomycin and lasalocid. This was expected, as monensin has the lowest $EC_{50}$ of the three ionophores *in vitro*, followed by salinomycin and lasalocid. Additionally, previously reported $LD_{50}$ values for rats indicate that monensin is more toxic to these animals than salinomycin and lasalocid [3].

The most significant ultrastructural effects were observed in the endomembrane system (including vacuoles, endosomes, lysosomes, endoplasmic reticulum, and Golgi apparatus), mitochondria, and the formation of autophagic vesicles. Affected myoblasts in both cell lines

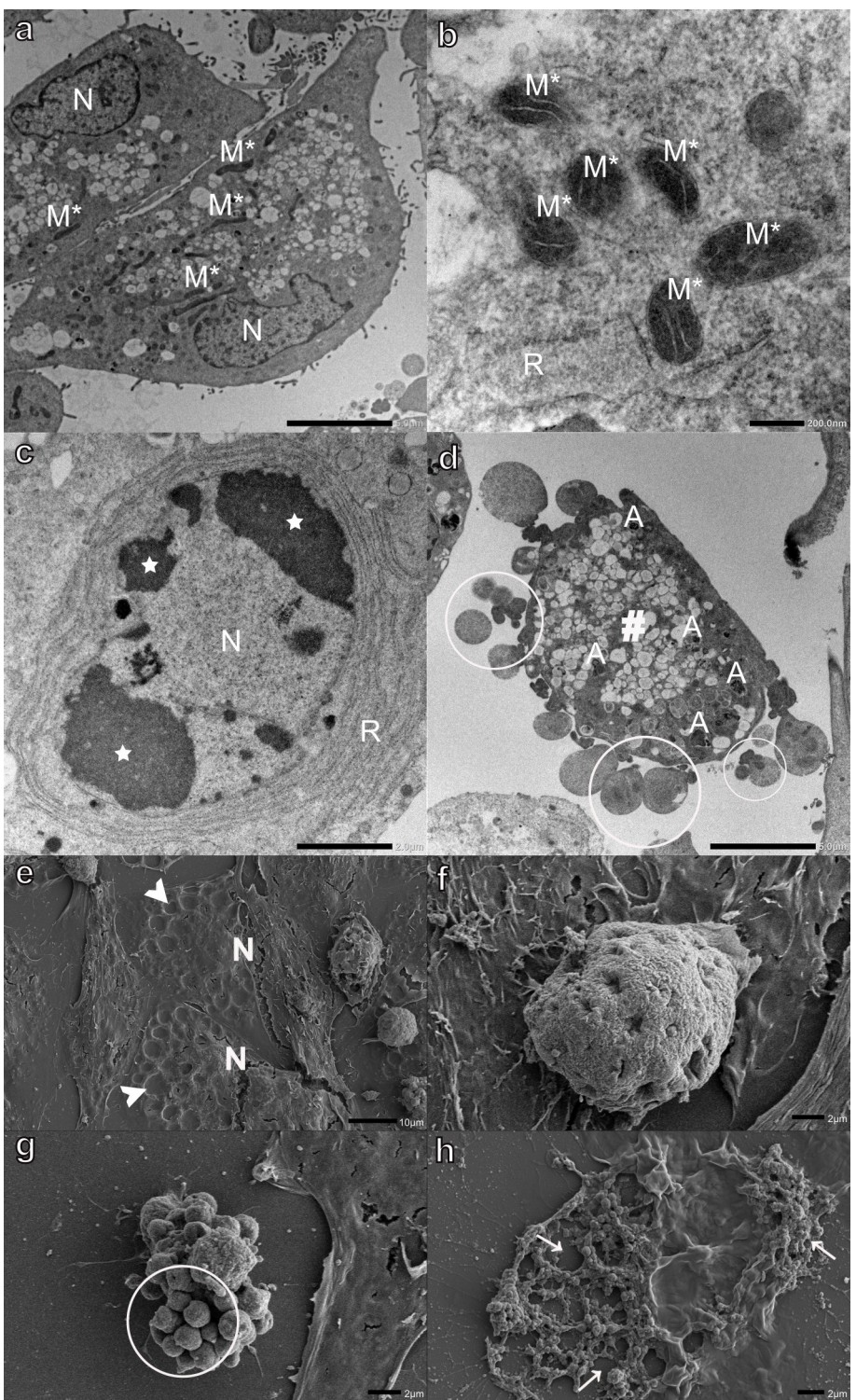

**Fig 6.** Transmission (a-d) and scanning (e-h) electron micrographs of L6 myoblasts exposed to 0.1 μM monensin for 48 h. Arrows-loss of membrane integrity, arrowheads-surface indentations, circled-apoptotic bodies, hash sign-condensed cytoplasm, stars-condensed chromatin, A-autophagic vesicles, M*-condensed mitochondria, N-nucleus, and R-rough endoplasmic reticulum. Scale bars: a = 200 nm, b = 1 μm, c, f-h = 2 μm, d = 5 μm, e = 10 μm.

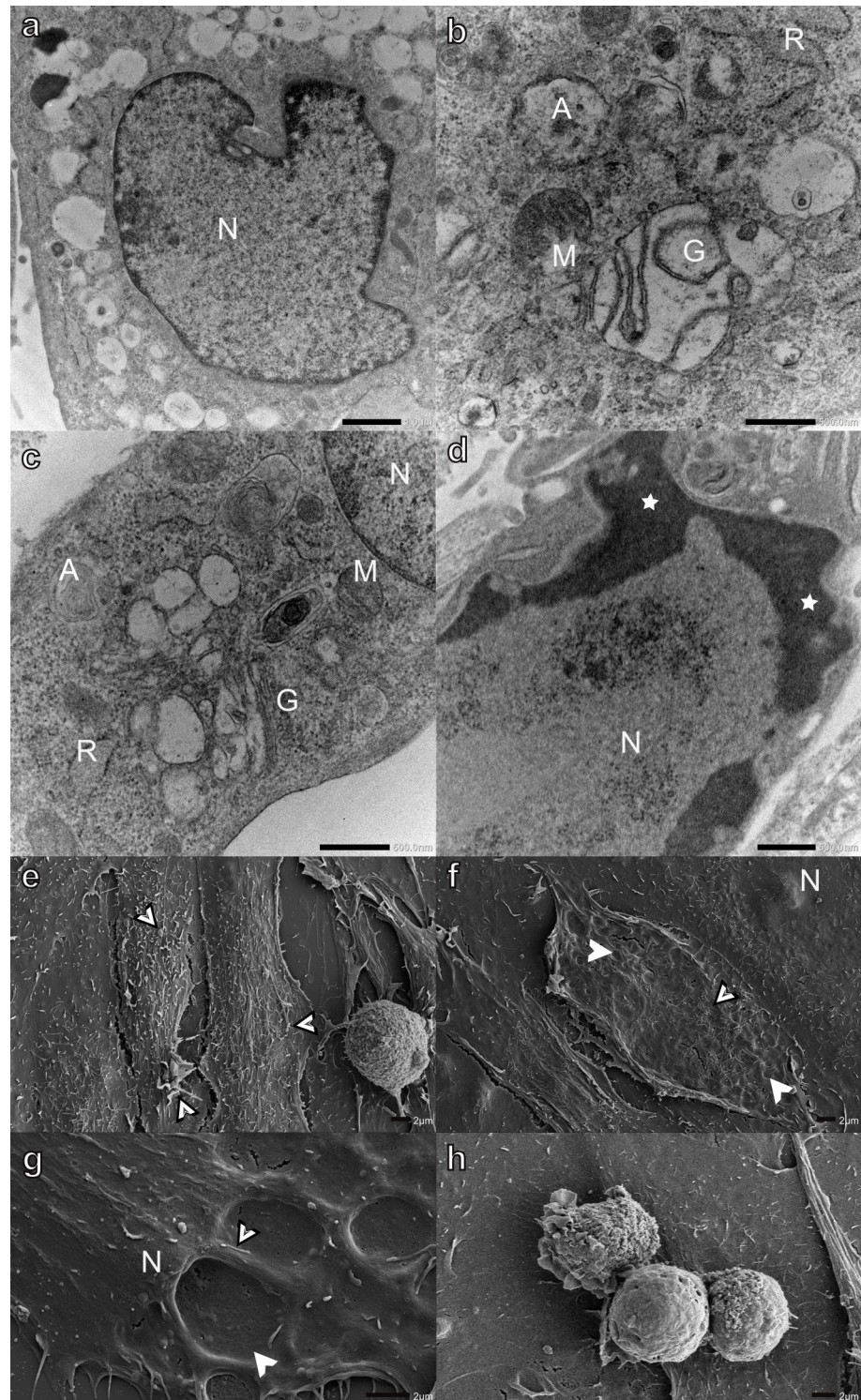

**Fig 7.** Transmission (a-d) and scanning (e-h) electron micrographs of L6 myoblasts exposed to 0.1 μM salinomycin for 48 h. Arrowheads-surface indentations, bordered arrowheads-filipodia-, bleb-, and frill-like surface structures, stars-condensed and marginalized chromatin, A-autophagic vesicles, G-Golgi apparatus, M-mitochondria, N-nucleus, and R-rough endoplasmic reticulum. Scale bars: a = 1 μm, b-d = 500 nm, e-h = 2 μm.

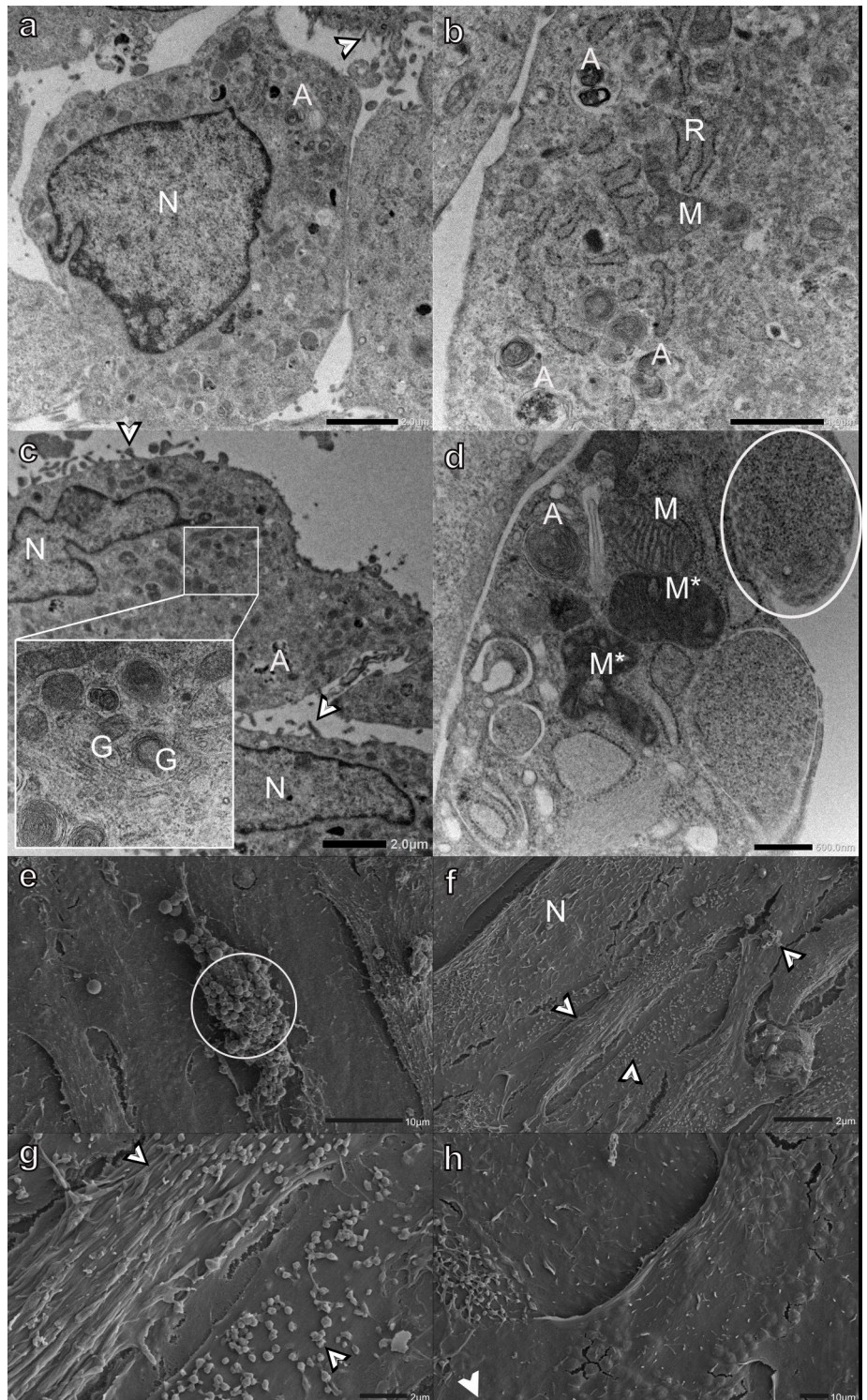

**Fig 8.** Transmission (a-d) and scanning (e-h) electron micrographs of L6 myoblasts exposed to 0.1 µM lasalocid for 48 h. Arrowheads-surface indentation, bordered arrowheads- filipodia-, bleb-, and frill-like surface structures, circled-apoptotic bodies, A-autophagic vesicles, G-Golgi apparatus, M-mitochondria, M*-condensed mitochondria, N-nucleus, and RER-rough endoplasmic reticulum. Scale bars: a, f-g = 2 µm, b-c = 1 µm, d = 500 nm, e, h = 10 µm.

exhibited extensive cytoplasmic vacuolization, which appeared as indentations on the myoblast surface, as observed by SEM. To our knowledge, this is the first study to use SEM to investigate the effects of ionophores on cardiac and skeletal myoblasts and to report that the extent of vacuolization causes visible alterations in the myoblast surface structure. Cytoplasmic vacuolization is a well-known morphological phenomenon, generally accepted as a physiological adaptive response to limit damage following a pathological stimulus [37, 38]. However, vacuolization can occur through several molecular mechanisms, involving diverse pathways and originating from different membrane sources and organelles [37, 38]. This study demonstrates that ionophore-induced physiological imbalance manifests ultrastructurally as swelling and vacuolization. This response is due to the disruption of the ionic balance across plasma and organelle membranes [3, 11–14, 39, 40], leading to changes in intracellular pH [3, 12, 19, 41], and consequently creating osmotic pressure gradients across organelle membranes [3, 11–14, 37, 38, 41–43]. Various organelles, including lysosomes, endosomes, the Golgi apparatus, and the endoplasmic reticulum, can be involved in forming vacuoles [44]. Furthermore, fusion of endomembrane vesicles, either osmotically induced, or to increase vacuolar membrane surface area in response to acidic ionophores, may additionally contribute to this vacuolization [38]. The Golgi apparatus is one of the first organelles affected after ionophore exposure. The Golgi apparatus in myoblasts exposed to monensin was dilated and swollen to such a degree that the organelle was not discernible as stacked cisternae, only as large vesicular profiles or electron-lucent vacuoles in the perinuclear space. Salinomycin or lasalocid exposure lead to swollen cisternae at the trans-face of the Golgi apparatus, resulting in vacuolization, frequently around the myonuclei. This phenomenon has been attributed to the inhibition of cellular transport at the trans-face of the Golgi apparatus [16].

Both cardiac and skeletal muscle myoblasts showed condensed mitochondria and mitochondrial degeneration and fragmentation, most prominently after monensin exposure. Ionophores disrupt normal ion homeostasis, leading to increased intracellular calcium concentrations [3, 11–13, 15–18]. Mitochondria buffer the excess intracellular calcium caused by the disrupted ion homeostasis, but their capacity becomes exhausted, calcium overload occurs, which then lead to mitochondrial damage and disrupted oxidative phosphorylation [11–13, 17, 19, 20, 45]. In comparative studies, experimental *in vitro* exposure to toxic doses of an ionophore led to condensed mitochondria with granular matrices in cardiac myofibers 24 to 48 hours post-exposure, progressing to mitochondrial swelling and fragmentation after 72 hours [22, 23]. Skeletal myofibers presented swollen mitochondria in fibers undergoing lysis, with a combination of swollen or condensed mitochondria in fibers with sub-lethal damage [24]. L6 myoblasts exposed to monensin contained abnormally elongated mitochondria, an observation not previously reported in this cell type. In freshwater algae and aquatic plants (*Micrasterias denticulata* and *Lemna* sp.), ionic stress induces the elongation and fusion of mitochondria, possibly as a mechanism to assist respiration and prevent stress-induced rupture of the mitochondrial outer membrane [46]. Thus, mitochondria in L6 myoblasts may fuse to maintain their function under ion imbalance caused by ionophore exposure, although further investigation is needed to confirm this.

Several myoblasts from both cell lines underwent cell death. Cell death pathways are traditionally classified into two broad types: programmed cell death (apoptosis) and non-programmed cell death (necrosis) [30, 31, 47–50]. Programmed cell death can further be subdivided into apoptotic and non-apoptotic phenotypes [30, 31, 47–50]. Initially, these pathways were considered mutually exclusive cellular states. However, recent research confirmed that, depending on the stimulus and cellular context, these pathways can interact in a cooperative or complementary manner [49, 51]. In the current investigation, several characteristic morphological features of apoptotic cell death were observed. TEM analysis revealed

chromatin condensation, loss of membrane surface structure, membrane blebbing, and apoptotic body formation. SEM analysis further substantiated the apoptotic morphotype, showing smooth membranes and extensive apoptotic body formation. Interestingly, more apoptotic myoblasts were observed after monensin exposure than after salinomycin or lasalocid exposure. Characteristic features of necrotic cell death (necrosis or necroptosis), such as cellular and organelle swelling, cytoplasmic degeneration, vacuolization, and plasma membrane disruption, were observed in both H9c2 and L6 myoblasts after ionophore exposure. Furthermore, autophagic vesicle formation, extensive cytoplasmic vacuolization, mitochondrial damage, ranging from condensation to fragmentation, and the formation of mitophagy-like structures were observed. These observations are suggestive of programmed non-apoptotic cell death mechanisms characterised by excessive vacuole formation (autophagy, methuosis, and paraptosis) and mitochondrial-dependent cell death (mitoptosis) [47].

Toxic ionophore exposure is associated with various cell death modalities induced by different cellular pathways and mechanisms. Programmed apoptotic cell death associated with oxidative stress and phosphorylation [18–20, 52, 53], reactive oxygen species (ROS) production [18–21, 53], alterations in the mitochondrial permeability transition (MPT) pore [18, 52, 54], mitochondrial hyperpolarization [20, 27, 28, 55], cell cycle arrest [27, 28], and/or caspase activation [52, 53, 56] have been reported [18–21, 27, 28, 53, 55, 56]. Necrosis has been observed associated with calcium overload [3, 57], MPT onset [58], mitochondrial dysfunction independent of the MPT [55], and/or reduced mitochondrial respiration [3, 11–13]. Simultaneous apoptosis and necrosis [21] or a predisposition to one modality has been linked to intracellular calcium involvement [59], and / or the dose of the ionophore [52, 60]. Several other non-apoptotic programmed cell death modalities have also been reported following ionophore exposure. Activation of canonical and noncanonical autophagy pathways [16, 19–21, 53], associated with impaired endolysosomal functions [61, 62], osmotic imbalances [41], ER stress [63–65], ROS production, and activation of protein kinase signalling has been shown.in numerous cell lines [16, 19–21, 41, 53, 61–65]. Föller et al. (2008) reported a non-apoptotic cell death; eryptosis, as a result of oxidative stress, ion channel activation, and membrane phospholipid scrambling [66]. Mitoptosis [53, 62], mitophagy, and mitochondria-mediated cell death [64, 67] resulting from the loss of membrane potential, reduced ATP, dose-dependent oxidative stress, and ROS production have been observed [53, 62, 64, 67]. Furthermore, the extensive cytoplasmic vacuolization, as observed in ionophore toxicosis, has been implicated in other non-apoptotic cell death modalities [68–70]. These include methuosis, resulting from dysfunctional ion regulation mechanisms [68], and paraptosis, after mitochondrial calcium overload, and decreased mitochondrial potential [69, 70]. Based on the morphological results of the H9c2 and L6 cell lines in this study, along with previous ultrastructural and biochemical investigations [11, 13, 14, 16, 18–28, 37, 38, 41–43, 53–70], it is evident that carboxylic ionophore exposure can lead to various cell death modalities, successive sequences, or convergent combinations of these mechanisms. This raises interesting questions about the precise mechanisms of carboxylic ionophores in cell death. Ultrastructural features alone are insufficient [48–50] and further studies using molecular techniques are needed to identify the specific cellular targets initiating cell death and the interplay between different cell death modalities.

In conclusion, carboxylic ionophores essentially alter ion gradients in myoblasts, with the initial morphological effects seen in the endomembrane system, most notably the Golgi apparatus, vacuolar system, and mitochondria. All three ionophores similarly altered the cellular ultrastructure, with monensin having the greatest effect at the concentrations investigated. Using this *in vitro* cardiac and skeletal muscle myoblast model, the ultrastructural observations correspond with some of the pathological changes, such as degeneration and necrosis, reported in the cardiac and skeletal muscles of livestock that have died from ionophore

toxicosis. While further research is needed to understand the molecular mechanisms and *in vivo* relevance, the ultrastructural changes and subsequent cell death observed in this study likely contribute to the lesions seen in animals exposed to lethal ionophore doses.

## Supporting information

**S1 Fig. Ionophore toxicity. (a-f)** Log-dose response curves were generated by using the mean percentage cell survival ± StEM vs the log of the concentration of the different ionophores (in µM). (Legend -●- 24 h, -■- 48 h, -▲- 72 h). **(g)** The EC50s (µM) ± StEM of the ionophores were exposed to three cell lines for 24, 48 and 72 h. n = number of biological repeats.
(TIFF)

**S2 Fig. Light microscopy images of ionophore cytotoxicity–Monensin.** H9c2 and L6 myoblasts exposed to 0.01, 0.1 and 1 µM monensin. Scale bar = 10 µm.
(TIFF)

**S3 Fig. Light microscopy images of ionophore cytotoxicity–Salinomycin.** H9c2 and L6 myoblasts exposed to 0.01, 0.1 and 1 µM salinomycin. Scale bar = 10 µm.
(TIFF)

**S4 Fig. Light microscopy images of ionophore cytotoxicity–Lasalocid.** H9c2 and L6 myoblasts were exposed to 0.01, 0.1, and 1 µM lasalocid. Scale bar = 10 µm.
(TIFF)

## Acknowledgments

We would like to thank Dr. James Wesley-Smith for allowing us to use the scanning electron microscope at the Electron Microscopy Unit of Sefako Makgatho Health Sciences University.

## Author Contributions

**Conceptualization:** Danielle Henn, Christo J. Botha.

**Data curation:** Antonia V. Lensink, Christo J. Botha.

**Formal analysis:** Danielle Henn, Antonia V. Lensink.

**Funding acquisition:** Danielle Henn, Christo J. Botha.

**Investigation:** Danielle Henn, Antonia V. Lensink.

**Methodology:** Danielle Henn, Antonia V. Lensink.

**Project administration:** Christo J. Botha.

**Resources:** Antonia V. Lensink, Christo J. Botha.

**Software:** Antonia V. Lensink.

**Supervision:** Christo J. Botha.

**Visualization:** Danielle Henn.

**Writing – original draft:** Danielle Henn.

**Writing – review & editing:** Antonia V. Lensink, Christo J. Botha.

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
