## [Decision Letter · Decision Letter 0]

8 May 2024

PONE-D-24-14459Ultrastructural changes in cardiac and skeletal myoblasts following in vitro exposure to monensin, salinomycin, and lasalocid.PLOS ONE

Dear Dr. Lensink,

Thank you for submitting your manuscript to PLOS ONE. After careful consideration, we feel that it has merit but does not fully meet PLOS ONE’s publication criteria as it currently stands. Therefore, we invite you to submit a revised version of the manuscript that addresses the points raised during the review process.

We have received detailed reviews from experts in your field. While both were generally positive about the study, they have expressed major concerns in numerous points. We hope that the expressed criticism and suggestions will help you to improve the concept and information presented in your manuscript and invite you to address all points raised by the reviewers,   Please submit your revised manuscript by Jun 22 2024 11:59PM. If you will need more time than this to complete your revisions, please reply to this message or contact the journal office at plosone@plos.org. Please include the following items when submitting your revised manuscript:A rebuttal letter that responds to each point raised by the academic editor and reviewer(s). You should upload this letter as a separate file labeled 'Response to Reviewers'.A marked-up copy of your manuscript that highlights changes made to the original version. You should upload this as a separate file labeled 'Revised Manuscript with Track Changes'.An unmarked version of your revised paper without tracked changes. You should upload this as a separate file labeled 'Manuscript'.

We look forward to receiving your revised manuscript.

Kind regards,

Hans-Peter Kubis, PD. Dr. rer. nat.

Academic Editor

PLOS ONE

Journal Requirements:

Reviewers' comments:

Reviewer's Responses to Questions

**Comments to the Author**

1. Is the manuscript technically sound, and do the data support the conclusions?

Reviewer #1: Partly

Reviewer #2: Partly

2. Has the statistical analysis been performed appropriately and rigorously? 

Reviewer #1: N/A

Reviewer #2: N/A

3. Have the authors made all data underlying the findings in their manuscript fully available?

Reviewer #1: Yes

Reviewer #2: Yes

4. Is the manuscript presented in an intelligible fashion and written in standard English?

Reviewer #1: Yes

Reviewer #2: Yes

5. Review Comments to the Author

Reviewer #1: Manuscript Number: PONE-D-24-14459

Article Type: Research Article

Title: Ultrastructural changes in cardiac and skeletal myoblasts following in vitro exposure to

monensin, salinomycin, and lasalocid.

Authors: Danielle Henn, Antonia Vergina Lensink, Christo J Botha

In this manuscript submitted to Plos One journal as Research Article, the authors, with some experience in studying effects of ionophores in myocytes, reported interesting ultrastructural data obtained in vitro on two rat muscle cell lines (cardiac H9c2 and skeletal L6 myoblasts) consecutive to their incubation for 48 h with a single dose of three ionophore molecules, monensin, salinomycin, and lasalocid. The paper is concise and well written; the language is clear, unambiguous and in general correct (apart from several grammatical errors, or inadequate formulations, mentioned below). The paper has a good structure: Introduction provides enough background to illustrate the importance of the topic, Methodology is described in enough detail to allow replication of the experiments, the Results are relatively well presented (presentation must be improved), and the whole paper is based on classic references. The topic of the manuscript is relevant to the field of the Plos One journal, fits with the scope of this journal and could be of interest for certain readers of this journal. However, before recommending the publication of this manuscript, this reviewer considers that a major revision is required. Thus, there are 3 major and 32 minor aspects to be solved:

Major points:

1. The TEM and especially SEM images are of low quality and details are difficult to observe or are not visible at all in printed form. If the authors could not provide better images, they should ask the editing team (if paper will be accepted) whether is possible to improve somehow the images. Please replace Fig 5 d. A cell with disrupted plasma membrane is not representative for a control group. Magnification bars in SEM images, cannot be observed, and in all TEM and SEM images the corresponding dimensions (in µm) cannot be read. The authors could insert magnification bars in each panel of SEM images (according to their size in original images) and mention the corresponding dimensions in the legends of all panels in TEM/SEM figures.

2. Apart from the aspects mentioned as “minor points” concerning the Discussion section, the authors have to compare in this section, and to discuss in more detail the results obtained for the two cell lines incubated with the three types of ionophores (and in relation with supplementary literature data). In this form of the manuscript, the authors only reported effects already known and mentioned into the scientific literature (by at least 9 papers cited here, and probably by many others that were not cited). Moreover, the authors have to point out the novelty of their approach in testing the toxicity of the three ionophores, the degree of originality of their results reported here, as well as the relevance of their original findings for the basic science and for the veterinary medicine (since not all the readers, including this reviewer are familiar with the problems raised by the cattle medication). They should also discuss the actuality of their work in a more general economic context (despite the fact that they did not use recent references – and if indeed there are no recent papers dealing with this topic, they should speculate the absence of such new data). Finally, the authors should discuss the extrapolation of their results to the clinical use of the tested ionophores.

3. Please rewrite entirely the conclusions, based on the obtained results (instead of repeating some of the results), with possible recommendation for the future use of these ionophores in the veterinary medicine.

Minor points (in the order of appearance):

1. Lines 6 and 18: “¶These authors contributed equally to this work.” And is only one author labeled with this symbol.

2. Line 30: in Abstract, please indicate the precise conditions of ionophores exposure (dose and time).

3. Line 39: Please be consistent with formatting of keywords.

4. Line 51: Please be more specific since not all ionophores have the same molecular mechanism and the same effect on cells.

5. Lines 51-54: please distribute the citations more accurately, since is difficult to find the enumerated effects into the 10 references listed at the end of the paragraph.

6. Lines 61-62. Please replace ”increased number of smooth endoplasmic reticulum and lipid droplets” with ”increased surface of smooth endoplasmic reticulum and number of lipid droplets”, since ER is a network of membrane within the cytoplasm. Alternatively, please use “number of ER profiles observed on the sections”.

7. Lines 89,90. If the authors cannot provide additional experimental data, please justify either here or in Discussion how the used dose of ionophores (0.1 µM) and the exposure time (48 h) were calculated (or selected) for incubation. For a better relevance of the result, at least two doses should be compared, or if a certain dose is more relevant, it should be tested for at least two different exposure times.

8. Line 135: “condensed mitochondria” are not visible in Fig. 2a-c. On the contrary, the presence of inner membranes and other debris in some of the electron lucent vacuoles is an indicative that mitochondria swelled and are responsible partially for the cytoplasmic vacuolation. This affirmation is supported by the round-shape of vesicles derived from mitochondria, while the expanded ER shows polymorphism. Moreover, this affirmation is also in line with the literature results cited in Introduction. Indeed, some condensed mitochondria are observed in Fig 2d, but this result is not mentioned in the main text.

9. Line 136. It is very difficult to recognize the “autophagic vesicles” – not labelled in the panels a-c (and to delineate them from secondary lysosomes appearing in high number) in Fig. 2, since the double or multiple membranes of such vesicles could not be identified at this low resolution. The same comment for the other figures.

10. Line 140. Please explain here or discuss in Discussion the “indentations”, since they are not visible by TEM at the periphery of cells. This reviewer suggests that they resulted from the collapse of the multiple vacuoles either during dehydration followed by drying instead of epoxy infiltration, or/and during sputter-coating process, when the cells were exposed to a high vacuum.

11. Line 144. The “apoptotic bodies” and “the red arrow” are not visible in Fig 2 g. It seems that authors inversed panels g and h of Fig 2 or explained inversely the results showed in the two panels.

12. Please rephrase line 146.

13. Line 148. Please indicate the panel(s) of Fig 3 showing the presence of autophagic vesicles and indicate these structures in the corresponding panel(s).

14. There is no “pink arrow” in Fig 4h.

15. Line 167. “A”, “G”, “M” and “RER” are not visible in the panels of Fig 2.

16. Line 171. “A” and “RER” are not visible in the panels of Fig 3. In Fig 3c “G” should be placed in the lower left side of the image.

17. Line 175. “pink arrows” and “A” are not visible in the panels of Fig 4.

18. Please rephrase line 180 (accord).

19. Line 192. Please indicate the panels in Fig 6 showing “autophagic vesicles”.

20. Line 194. “Red asterisks” are not visible in Fig 6.

21. Lines 190-196. Fig 6c is neither presented nor cited into the text.

22. Lines 200-201. Fig 6g and 6h are inversely explained.

23. Lines 202 and 205. The authors should indicate here or discuss later the origin of vesicles (numerous as seen in Fig 7a), since “mitochondria and RER were unaffected”.

24. Lines 206-208. Please check the correspondence of the presented results with the panels e-h of Fig 7.

25. Line 220. “A” is not visible in the panels of Fig 5.

26. Line 224. “red asterisks” are not visible in the panels of Fig 6. Please explain white asterisks.

27. Line 228. “A” is not visible in the panels of Fig 7.

28. Line 233. “The pink arrow” in Fig 8h is of the same colour as the “red arrow” in Fig 8d.

29. Line 234. “A” is not visible in the panels of Fig 8.

30. Please remove all references to figures from Discussion section.

31. Please rephrase line 289 (accord).

32. Line 301. Please replace “method” with “type” or “process”.

Reviewer #2: PLOS ONE Manuscript #: PONE-D-24-14459

Title: Ultrastructural changes in cardiac and skeletal myoblasts following in vitro exposure to monensin, salinomycin, and lasalocid.

Authors: Danielle Henn; Antonia Vergina Lensink; Christo J Botha

The authors investigated the effects of three carboxylic ionophores that are used as antibiotics in animal agriculture on ultrastructure of cardiac (H9c2) and skeletal muscle (L6) myoblasts using transmission and scanning electron microscopy. Ionophore exposure resulted in condensed mitochondria, dilated Golgi apparatus, and electron-lucent vesicles distributed throughout the cytoplasm. Apoptotic and necrotic myoblasts were observed after ionophore exposure. The compound Monensin had the largest effects on ultrastructure of cardiac and skeletal muscle myoblasts.

Although the manuscript provides interesting information on the effect of carboxylic ionophores on ultrastructure of cardiac and skeletal muscle myoblasts that deserve to be considered for publication in PlosOne, several issues need to be addressed.

Major points

Why do the authors restrict their analysis to myoblasts instead of investigating the effects of the ionophores also in myotubes?

What is the rationale for using 0.1 �m for all three ionophores? Get treated animals actually the same dose of all three ionophores?

An Ethics Statement is missing in the Methods section.

Lines 300-302: Concerning distinction of ionophore-induced apoptosis and necrosis, the authors state that “morphological criteria alone is not sufficient”. In accordance, to assess myoblast apoptosis, annexin V staining should be considered to evaluate effects of ionophore treatment. Levels of lactate dehydrogenase (LDH), a marker of necrosis, should be determined in the supernatant of ionophore-treated myoblasts.

Several mismatches and inconsistencies between Results, Figures and Figure Legends need to be revised thoroughly:

Lines 135-136: “Condensed mitochondria (Fig 2 a-c)” are shown in Figure 2d.

Lines 136, 148, 155-156, and 192-193: How are “autophagic vesicles“ defined ultrastructurally and distinguished from other vesicles? Please indicate them in Figures 2a-c, 3a-c, 4, and Figure 6. Concerning autophagic vesicles, see also comments on Figure Legends below.

Lines 138, 143, 170, 196, 224, and 232: Which features define the budding structures as “apoptotic bodies“?

Lines 138-139, line 144: Please indicate features of necrosis like cellular debris in Figures 2d and h.

Lines 143-144: A red arrow supposed to mark apoptotic bodies in Figure 2g is shown in Figure 2h.

Line 160. The pink arrow supposed to mark indentations is missing in Figure 4h.

Legends Figures 1 to 4: several features listed in the Legends are not marked in the corresponding Figures. Figure 1: RER; Figures 2 and 3: A-autophagic vesicles, RER; Figure 4: A-autophagic vesicles.

Line 194. What means “marginalized chromatin? White but not red asterisks (see line 224) are shown in Figure 6b and d, obviously indicating mitochondria but not “condensed chromatin” (line 223-224).

Line 195. Please indicate “condensed cytoplasm” in Figure 6.

Line 197-198. Please rephrase: “…resulted in the surface of many myoblasts containing deep indentations scattered around the nucleus…”.

Line 200: Please specify “lacked surface structures”.

Line 201: Please indicate features of necrosis in Figure 6h.

Legends Figures 5 to 8: several features listed in the Legends are not marked in the corresponding Figures. Figure 6: A-autophagic vesicles, RER, red asterisks supposed to mark condensed (line 223-224) chromatin are white and indeed mark mitochondria, and h is supposed to show necrotic myoblasts (line 201), while the red arrow is supposed to show apoptotic bodies (line 224); Figure 7: A-autophagic vesicles; Figure 8: A-autophagic vesicles, a pink arrow supposed to show indentations (lines 214-215) is not shown in Figure 8g.

Please provide a conclusion based on your observations.

Minor points

Lines 130-133, and lines 187-189. These summaries should preferably appear at the end of respective paragraphs.

Lines 55-58: Concerning ionophore toxicosis, the authors argue about “a variable inflammatory component”. What are the ultrastructural features of inflammation? Are signs of inflammation observed in the myoblasts?

Lines 243-244: “Monensin blocks transport at the trans-face of the Golgi apparatus… .” Is this statement a conclusion from the data? Otherwise, please provide a reference.

Lines 248-249: What is the evidence that electron-lucent vesicles filling the entire myoblasts are the result of dilatation of the Golgi apparatus?

In this context:

What is the evidence for statements relating intracellular vesicles “distributed throughout the cytoplasm” (line 31) to indentations? Lines 31-32: “…electron-lucent vesicles distributed throughout the cytoplasm, which appeared as indentations on the myoblast surface.” Similar: Lines 240-241: “…accumulation of electron-lucent vesicles within the cytoplasm of the affected myoblasts, which manifested as indentations… “. Lines 249-250: “Vesiculation of the cytoplasm results in indentations on the myoblast surface, giving the myoblasts a pockmarked appearance.” Lines 304-305: “The vesicles manifest as indentations on the surface of the myoblasts.”

Lines 283-284: Please rephrase and specify “surface details”.

Lines 277-279, and 287-288. The characteristic morphological features of apoptosis and necrosis should be introduced in the Introduction. Please provide a reference for apoptotic features.

Line 295: Please rephrase “…. can drive a cell to favor necrosis over apoptosis.”

6. PLOS authors have the option to publish the peer review history of their article (what does this mean?). If published, this will include your full peer review and any attached files.

Reviewer #1: No

Reviewer #2: No

---

## [Author Response · Author response to Decision Letter 0]

21 Jun 2024

We are pleased to receive your communication on May 8, 2024, indicating a “Major revision” of our manuscript. The Editor and Reviewer’s comments were highly insightful and enabled us to improve the quality of the manuscript. I am pleased to submit the revised version of our manuscript. 

We have revised the manuscript based on the editor and reviewer’s comments, observations, and suggestions to the best of our abilities. 

We would like to thank you and the reviewers for your work in reviewing the manuscript and bringing up constructive inputs that will add value to the manuscript.

Our responses and answers to the editor and reviewer’s comments are hereby presented below.

Please if there is any other thing that needs to be clarified, we will appreciate it if we can be given another opportunity to do so.

Kind regards,

Dr AV Lensink 

Manager, Electron Microscope Unit

Reviewer #1: Manuscript Number: PONE-D-24-14459

In this manuscript submitted to Plos One journal as Research Article, the authors, with some experience in studying effects of ionophores in myocytes, reported interesting ultrastructural data obtained in vitro on two rat muscle cell lines (cardiac H9c2 and skeletal L6 myoblasts) consecutive to their incubation for 48 h with a single dose of three ionophore molecules, monensin, salinomycin, and lasalocid. The paper is concise and well written; the language is clear, unambiguous and in general correct (apart from several grammatical errors, or inadequate formulations, mentioned below). The paper has a good structure: Introduction provides enough background to illustrate the importance of the topic, Methodology is described in enough detail to allow replication of the experiments, the Results are relatively well presented (presentation must be improved), and the whole paper is based on classic references. The topic of the manuscript is relevant to the field of the Plos One journal, fits with the scope of this journal and could be of interest for certain readers of this journal. However, before recommending the publication of this manuscript, this reviewer considers that a major revision is required. Thus, there are 3 major and 32 minor aspects to be solved:

Major points:

1. The TEM and especially SEM images are of low quality and details are difficult to observe or are not visible at all in printed form. If the authors could not provide better images, they should ask the editing team (if paper will be accepted) whether is possible to improve somehow the images. Please replace Fig 5 d. A cell with disrupted plasma membrane is not representative for a control group. Magnification bars in SEM images, cannot be observed, and in all TEM and SEM images the corresponding dimensions (in µm) cannot be read. The authors could insert magnification bars in each panel of SEM images (according to their size in original images) and mention the corresponding dimensions in the legends of all panels in TEM/SEM figures.

Thank you for bringing this to our attention. The figures were redone using different software which allows more precise control of exporting the images with specific size and resolution parameters. The PDF generated from the submission also significantly decreased the brightness and resolution – we will ask the editing team if it can be improved in the publication (if the paper is accepted). 

Fig 5 d has been replaced with a more representative image for a control group sample.

New scale bars were added to the SEM images and scale bar dimensions were added to the legends of all panels.

2. Apart from the aspects mentioned as “minor points” concerning the Discussion section, the authors have to compare in this section, and to discuss in more detail the results obtained for the two cell lines incubated with the three types of ionophores (and in relation with supplementary literature data). In this form of the manuscript, the authors only reported effects already known and mentioned into the scientific literature (by at least 9 papers cited here, and probably by many others that were not cited). Moreover, the authors have to point out the novelty of their approach in testing the toxicity of the three ionophores, the degree of originality of their results reported here, as well as the relevance of their original findings for the basic science and for the veterinary medicine (since not all the readers, including this reviewer are familiar with the problems raised by the cattle medication). They should also discuss the actuality of their work in a more general economic context (despite the fact that they did not use recent references – and if indeed there are no recent papers dealing with this topic, they should speculate the absence of such new data). Finally, the authors should discuss the extrapolation of their results to the clinical use of the tested ionophores.

Thank you for the suggestions. We have elaborated on our discussion by making a more detailed comparison of the ultrastructural changes between the two cell lines, discussing previous findings as well as highlighting the novel factors of our approach. In our conclusion, we addressed the relevance of our data to basic/clinical research. 

Of note, there is a general lack of recent literature published discussing ionophores (outside of cancer research).

3. Please rewrite entirely the conclusions, based on the obtained results (instead of repeating some of the results), with possible recommendation for the future use of these ionophores in the veterinary medicine.

As requested, we have rewritten the conclusion. Although this study does not focus on recommendations for future use of ionophores in veterinary medicine, it does contribute to our understanding of how pathological lesions develop in cardiac and muscle tissue following exposure to high ionophore concentrations.

Minor points (in the order of appearance):

1. Lines 6 and 18: “¶These authors contributed equally to this work.” And is only one author labeled with this symbol.

The superscript was changed to indicate the first two authors contributed equally.

2. Line 30: in Abstract, please indicate the precise conditions of ionophores exposure (dose and time).

The information on dose and exposure time was added.

3. Line 39: Please be consistent with formatting of keywords.

The keyword formatting was corrected.

4. Line 51: Please be more specific since not all ionophores have the same molecular mechanism and the same effect on cells.

We elaborated and specifically indicated carboxylic ionophores that form zwitterionic complexes, to distinguish them from neutral ionophores which form charged complexes and have a considerably greater toxic effect. We also clarified that the detrimental effect may occur, further mentioning in the next sentence that high levels of ionophores produce toxic effects.

5. Lines 51-54: please distribute the citations more accurately, since is difficult to find the enumerated effects into the 10 references listed at the end of the paragraph.

The citations were distributed to reflect where each cited effect was found in the literature.

6. Lines 61-62. Please replace ”increased number of smooth endoplasmic reticulum and lipid droplets” with ”increased surface of smooth endoplasmic reticulum and number of lipid droplets”, since ER is a network of membrane within the cytoplasm. Alternatively, please use “number of ER profiles observed on the sections”.

We replaced ”increased number of smooth endoplasmic reticulum and lipid droplets” with” increased surface of smooth endoplasmic reticulum and several lipid droplets”.

7. Lines 89,90. If the authors cannot provide additional experimental data, please justify either here or in Discussion how the used dose of ionophores (0.1 µM) and the exposure time (48 h) were calculated (or selected) for incubation. For a better relevance of the result, at least two doses should be compared, or if a certain dose is more relevant, it should be tested for at least two different exposure times.

Due to time and budget constraints, we had to limit our study to using only one concentration. Additionally, we have carried out in vitro cytotoxicity experiments to determine the EC50 of the different ionophores. The concentration of 0.1 uM was chosen as it was a concentration that was below the EC50 of both Lasalocid and Salinomycin, but higher than that for Monensin. Thus, it demonstrates the different degrees by which the ionophores can affect cardiac and skeletal myoblasts at the same concentration. We have added additional supplementary images in this regard.

8. Line 135: “condensed mitochondria” are not visible in Fig. 2a-c. On the contrary, the presence of inner membranes and other debris in some of the electron lucent vacuoles is an indicative that mitochondria swelled and are responsible partially for the cytoplasmic vacuolation. This affirmation is supported by the round-shape of vesicles derived from mitochondria, while the expanded ER shows polymorphism. Moreover, this affirmation is also in line with the literature results cited in Introduction. Indeed, some condensed mitochondria are observed in Fig 2d, but this result is not mentioned in the main text.

Thank you for the suggestion, this has been added and will improve the manuscript. The electron-lucent vacuoles are seen in Fig 2 a-c, and the condensed and swollen/fragmented mitochondria in Fig 2 b, the features have been labelled and referred to in the text.

9. Line 136. It is very difficult to recognize the “autophagic vesicles” – not labelled in the panels a-c (and to delineate them from secondary lysosomes appearing in high number) in Fig. 2, since the double or multiple membranes of such vesicles could not be identified at this low resolution. The same comment for the other figures.

The autophagic vesicles were labelled in Figures 2a and 2c and a higher magnification inset was added to Figure 2c.

10. Line 140. Please explain here or discuss in Discussion the “indentations”, since they are not visible by TEM at the periphery of cells. This reviewer suggests that they resulted from the collapse of the multiple vacuoles either during dehydration followed by drying instead of epoxy infiltration, or/and during sputter-coating process, when the cells were exposed to a high vacuum.

We agree with your suggestion and thank you for pointing this out. We have added this explanation.

11. Line 144. The “apoptotic bodies” and “the red arrow” are not visible in Fig 2 g. It seems that authors inversed panels g and h of Fig 2 or explained inversely the results showed in the two panels.

The image sequence and labels were corrected. Furthermore, we changed the image labels to be consistent throughout the manuscript (e.g. apoptotic bodies = circled, arrowheads = surface indentations, stars = condensed and marginalized chromatin, etc.).

12. Please rephrase line 146.

The sentence was rephrased and changed from ”Salinomycin had less of an effect on myoblasts than monensin” to “Salinomycin affected myoblasts to a lesser degree when compared with the monensin-exposed cells”.

13. Line 148. Please indicate the panel(s) of Fig 3 showing the presence of autophagic vesicles and indicate these structures in the corresponding panel(s).

The panel showing autophagic vesicles was referred to in the text and the corresponding panel was correctly labelled.

14. There is no “pink arrow” in Fig 4h.

Labels were added to Fig 4 h, please note that we discarded the pink arrows and consistently replaced the labels of all indicated surface indentations in the manuscript with arrowheads.

15. Line 167. “A”, “G”, “M” and “RER” are not visible in the panels of Fig 2.

Labels “A”, “M*” and “R” were added to the panels, no “G” was shown in this panel.

16. Line 171. “A” and “RER” are not visible in the panels of Fig 3. In Fig 3c “G” should be placed in the lower left side of the image.

The correct labels were added and G indicated on the lower left of Fig 3c

17. Line 175. “pink arrows” and “A” are not visible in the panels of Fig 4.

Labels were added to Fig 4 h, please note that we discarded the pink arrows and consistently replaced the labels of all indicated surface indentations in the manuscript with arrowheads. A – autophagic vesicles were added to Fig 4 a and c.

18. Please rephrase line 180 (accord).

The sentence was rephrased and changed from ”A large nucleus was generally located near the center of the myoblast, with one or more Golgi apparatus in close proximity” to “Myonuclei was typically centrally positioned, often with one or more adjacent Golgi apparatus visible”.

19. Line 192. Please indicate the panels in Fig 6 showing “autophagic vesicles”.

A label and the relevant in-text reference were added.

20. Line 194. “Red asterisks” are not visible in Fig 6.

The correct label was added to Figure 6c, please note that we discarded the red asterisk and consistently replaced the labels in all the figures showing condensed and marginalized chromatin with stars.

21. Lines 190-196. Fig 6c is neither presented nor cited into the text.

A reference to Fig 6c was added to the text. 

22. Lines 200-201. Fig 6g and 6h are inversely explained.

The image sequence was corrected.

23. Lines 202 and 205. The authors should indicate here or discuss later the origin of vesicles (numerous as seen in Fig 7a), since “mitochondria and RER were unaffected”.

We have elaborated on the origin of the vesicles/vacuoles in the Discussion section

24. Lines 206-208. Please check the correspondence of the presented results with the panels e-h of Fig 7.

The image sequence was corrected, labels were added to the panels, and in-text references were corrected.

25. Line 220. “A” is not visible in the panels of Fig 5.

The correct label was added.

26. Line 224. “red asterisks” are not visible in the panels of Fig 6. Please explain white asterisks.

The label was added to Figure 6c, please note that we discarded the red asterisk and consistently replaced the labels of all the figures showing condensed and marginalized chromatin with stars. The white asterisk indicated the condensed mitochondria, however, this label was discarded and consistently replaced with “M*”, this was also added to the figure legend.

27. Line 228. “A” is not visible in the panels of Fig 7.

The correct label was added.

28. Line 233. “The pink arrow” in Fig 8h is of the same colour as the “red arrow” in Fig 8d.

The relevant labels were corrected, please note that we discarded the coloured arrows and consistently labelled all apoptotic bodies in the manuscript by circling them, and indicating the surface indentations with arrowheads. 

29. Line 234. “A” is not visible in the panels of Fig 8.

The correct label was added.

30. Please remove all references to figures from Discussion section.

All the references to figures were removed from the Discussion section.

31. Please rephrase line 289 (accord).

The sentence was rephrased and changed from ”After ionophore exposure, only the Golgi apparatus of both H9c2 and L6 myoblasts were swollen, while the mitochondria were condensed, and the RER remained unaffected” to “After ionophore exposure, the Golgi apparatus in both H9c2 and L6 myoblasts displayed swelling, whereas the mitochondria appeared condensed, with no discernible effect observed on the rough endoplasmic reticula”.

32. Line 301. Please replace “method” with “type” or “process”.

As requested, we replaced “method” with “type”.

Reviewer #2: PLOS ONE Manuscript #: PONE-D-24-14459

The authors investigated the effects of three carboxylic ionophores that are used as antibiotics in animal agriculture on ultrastructure of cardiac (H9c2) and skeletal muscle (L6) myoblasts using transmission and scanning electron microscopy. Ionophore exposure resulted in condensed mitochondria, dilated Golgi apparatus, and electron-lucent vesicles distributed throughout the cytoplasm. Apoptotic and necrotic myoblasts were observed after ionophore exposure. The compound Monensin had the largest effects on ultrastructure of cardiac and skeletal muscle myoblasts.

Although the manuscript provides interesting information on the effect of carboxylic ionophores on ultrastructure of cardiac and skeletal muscle myoblasts that deserve to 

---

## [Decision Letter · Decision Letter 1]

5 Jul 2024

PONE-D-24-14459R1Ultrastructural changes in cardiac and skeletal myoblasts following in vitro exposure to monensin, salinomycin, and lasalocid.PLOS ONE

Dear Dr. Lensink,

Thank you for submitting your manuscript to PLOS ONE. After careful consideration, we feel that it has merit but does not fully meet PLOS ONE’s publication criteria as it currently stands. Therefore, we invite you to submit a revised version of the manuscript that addresses the points raised during the review process.

We have now received the reviews for your R1 version of your manuscript. While one reviewer found this version acceptable, the other reviewer suggests further alterations of your manuscript. I am sure that the comments will lead to a further improvement of your manuscript and I am looking forward to your responses and alterations for this work. The reviewer is an established expert in your field and I am grateful that the reviewer is investing so much time and effort for improving the clarity of evidence for your work.  Please submit your revised manuscript by Aug 19 2024 11:59PM. If you will need more time than this to complete your revisions, please reply to this message or contact the journal office at plosone@plos.org. Please include the following items when submitting your revised manuscript:A rebuttal letter that responds to each point raised by the academic editor and reviewer(s). You should upload this letter as a separate file labeled 'Response to Reviewers'.A marked-up copy of your manuscript that highlights changes made to the original version. You should upload this as a separate file labeled 'Revised Manuscript with Track Changes'.An unmarked version of your revised paper without tracked changes. You should upload this as a separate file labeled 'Manuscript'.If applicable, we recommend that you deposit your laboratory protocols in protocols.io to enhance the reproducibility of your results. Protocols.io assigns your protocol its own identifier (DOI) so that it can be cited independently in the future. For instructions see: https://journals.plos.org/plosone/s/submission-guidelines#loc-laboratory-protocols. Additionally, PLOS ONE offers an option for publishing peer-reviewed Lab Protocol articles, which describe protocols hosted on protocols.io. Read more information on sharing protocols at https://plos.org/protocols?utm_medium=editorial-email&utm_source=authorletters&utm_campaign=protocols.

We look forward to receiving your revised manuscript.

Kind regards,

Hans-Peter Kubis, PD. Dr. rer. nat.

Academic Editor

PLOS ONE

Journal Requirements:

Reviewers' comments:

Reviewer's Responses to Questions

**Comments to the Author**

1. If the authors have adequately addressed your comments raised in a previous round of review and you feel that this manuscript is now acceptable for publication, you may indicate that here to bypass the “Comments to the Author” section, enter your conflict of interest statement in the “Confidential to Editor” section, and submit your "Accept" recommendation.

Reviewer #1: All comments have been addressed

Reviewer #2: (No Response)

2. Is the manuscript technically sound, and do the data support the conclusions?

Reviewer #1: Yes

Reviewer #2: Yes

3. Has the statistical analysis been performed appropriately and rigorously? 

Reviewer #1: N/A

Reviewer #2: N/A

4. Have the authors made all data underlying the findings in their manuscript fully available?

Reviewer #1: Yes

Reviewer #2: Yes

5. Is the manuscript presented in an intelligible fashion and written in standard English?

Reviewer #1: Yes

Reviewer #2: No

6. Review Comments to the Author

Reviewer #1: Congratulations for the revised form of your manuscript.

Congratulations for the revised form of your manuscript.

Reviewer #2: PLOS ONE Manuscript #: PONE-D-24-14459R1

Title: Ultrastructural changes in cardiac and skeletal myoblasts following in vitro exposure to monensin, salinomycin, and lasalocid.

Authors: Danielle Henn; Antonia Vergina Lensink; Christo J Botha

The authors investigated the effects of three carboxylic ionophores that are used as antibiotics in animal agriculture on ultrastructure of cardiac (H9c2) and skeletal muscle (L6) myoblasts using transmission and scanning electron microscopy. Ionophore exposure resulted in condensed mitochondria, dilated Golgi apparatus, and electron-lucent vesicles distributed throughout the cytoplasm. Apoptotic and necrotic myoblasts were observed after ionophore exposure. The compound Monensin had the largest effects on ultrastructure of cardiac and skeletal muscle myoblasts.

Although the revised manuscript is significantly improved in response to the reviewer’s comments, again several issues especially related to Results, Figures and Figure Legends need to be addressed.

Major points

Line 167 [Numbers from manuscript with Track Changes]: In Figure 2b, an inset supposed to demonstrate “mitophagy-like structures” is missing.

Line 168: What does “mitochondrial vacuolar degeneration” mean? The structures shown in Figure 2b related to category M° (lines 222-223), supposed to show “mitochondrial vacuolar degeneration”, are not convincing.

Lines 172-173: Features of autophagic vesicles (“double or multiple membranes”) are not convincingly demonstrated in Figure 2c, inset.

Figure 192-3: Especially in Figure 3b and c, it is not clear why structures seemingly representing electron-lucent vacuoles are assigned as dilated Golgi apparatus.

Lines 214-215: It is not clear, what the bordered arrowheads are supposed to show in Figure 1e-1h. Line 156-157 indicate “filipodia-like and bleb-like surface structures” in Figure 1h that should be assigned.

Lines 227-228: Some of the arrowheads in Figure 3e clearly do not show surface indentations. In the results related to Figure 3, indentations are not covered at all as a feature of H9c2 exposed to salinomycin.

Lines 253-254: At least Figure 6a seems not to show “myoblasts … filled with electron-lucent vacuoles…”.

Lines 270-271: When comparing Figure 7a with Figure 6a and b, it seems not to be justified to state that “Myoblasts exposed to salinomycin were less vacuolated than those

exposed to monensin (Fig 7 a)…”.

Line 314: The structure encircled in Figure 8e is supposed to show apoptotic bodies, that seem to have a quiet different structure as compared with the apoptotic bodies shown in Figure 2g, 3g, and 6g. Please comment.

Line 316: Golgi apparatus seems hardly to be visible in Figure 8c.

Lines 351-354: An effect of monenesin on the Golgi apparatus is neither demonstrated in Figure 2 for H9c2 myoblasts nor in Figure 6 for L6 myoblasts.

The discussion, while in general improved and much more interesting, is now a little bit lengthy and not always very stringent.

Minor points

Line 151: should read: “in addition to” instead of “with some”.

Line 170, line 362: Please delete “with”.

Line 188-189: Please rephrase “…necrotic myoblasts were observed with little

to no membrane continuity, mostly only comprising cellular debris…”

Lines 205 and 235: Please specify, what kind of surface structures are indicated by bordered arrowheads in Figures 4e-g.

Line 214: Why are H9c2 myoblasts incubated in DMEM for 48 h termed as “negative control”?

Line 228: Should read “filipodia- and bleb-like surface structures” (line 199) instead of “surface structures”.

Lines 220, 228, 300, and 314: Should read “encircled “.

Lines 306-307: Arrowheads are supposed to show indentations. Concerning Figure 7f, this statement is questionable.

Lines 307: Please specify, what kind of structures the bordered arrowheads are supposed to show.

Line 356: Please rephrase “The number of endoplasmic reticulum profiles also increased…”.

Line 404. Please rephrase “vacuole-presenting cell death”.

7. PLOS authors have the option to publish the peer review history of their article (what does this mean?). If published, this will include your full peer review and any attached files.

Reviewer #1: No

Reviewer #2: No

---

## [Author Response · Author response to Decision Letter 1]

23 Aug 2024

Dear Editorial Team and Reviewer

Thank you for the time and effort invested in the review of this manuscript, your comments and suggestions were insightful and constructive, we believe the changes suggested improves the quality and clarity of evidence in this work. 

A financial disclosure and a role of funder statement has been added to our cover letter as requested from the editorial office, and the reviewers comments has been addressed, please see responses below.

Best regards

Antoinette Lensink

Reviewer #2: PLOS ONE Manuscript #: PONE-D-24-14459R1 

Title: Ultrastructural changes in cardiac and skeletal myoblasts following in vitro exposure to monensin, salinomycin, and lasalocid. 

Authors: Danielle Henn; Antonia Vergina Lensink; Christo J Botha 

The authors investigated the effects of three carboxylic ionophores that are used as antibiotics in animal agriculture on ultrastructure of cardiac (H9c2) and skeletal muscle (L6) myoblasts using transmission and scanning electron microscopy. Ionophore exposure resulted in condensed mitochondria, dilated Golgi apparatus, and electron-lucent vesicles distributed throughout the cytoplasm. Apoptotic and necrotic myoblasts were observed after ionophore exposure. The compound Monensin had the largest effects on ultrastructure of cardiac and skeletal muscle myoblasts. 

Although the revised manuscript is significantly improved in response to the reviewer’s comments, again several issues especially related to Results, Figures and Figure Legends need to be addressed.

Major points 

Line 167 [Numbers from manuscript with Track Changes]: In Figure 2b, an inset supposed to demonstrate “mitophagy-like structures” is missing.

 - Thank you for pointing out this mistake, the correct figure has been uploaded.

Line 168: What does “mitochondrial vacuolar degeneration” mean? The structures shown in Figure 2b related to category M° (lines 222-223), supposed to show “mitochondrial vacuolar degeneration”, are not convincing.

Mitochondrial vacuolar degeneration has been described as the derangement of mitochondrial structural integrity which can be morphologically characterised by progressive stages. Initially the mitochondrial cristae undergo conformational changes, followed by focal loss and dissolution of the cristae. Advancing stages show the multifocal loss of the cristae, mitochondrial swelling, significant to complete dissolution of the cristae, permeabilization and loss of the inner mitochondrial membrane and eventually the rupture of the outer mitochondrial membrane. This is a common aetiology of oxidative stress, ROS production and increased mitochondrial calcium which also results from toxic ionophore exposure. 

 - Upon the recommendation of another reviewer: “…presence of inner membranes and other debris in some of the electron lucent vacuoles is an indicative that mitochondria swelled and are responsible partially for the cytoplasmic vacuolation. This affirmation is supported by the round-shape of vesicles derived from mitochondria…”, and further comparison of micrographs depicting mitochondria undergoing vacuolar degeneration in literature (references 1- 5 below) we decided to include this terminology to highlight the degree of mitochondrial damage observed. However, we have rewritten the relevant section, adding a further morphological description and changing the word choices to convey the fact that these structures is suggestive of mitochondrial vacuolar degeneration.

1. Chaanine AH. Morphological Stages of Mitochondrial Vacuolar Degeneration in Phenylephrine-Stressed Cardiac Myocytes and in Animal Models and Human Heart Failure. Medicina (Kaunas). 2019. 55(6):239. 

2. Chaanine AH, LeJemtel T, Delafontaine P. Mitochondrial Pathobiology and Metabolic Remodeling in Progression to Overt Systolic Heart Failure. Journal of Clinical Medicine. 2020. 9. 3582. 

3. Kong J, Xu Z. Massive mitochondrial degeneration in motor neurons triggers the onset of amyotrophic lateral sclerosis in mice expressing a mutant SOD1. J Neurosci. 1998. 18(9):3241-50. 

4. Aliev G, Li Y, Palacios HH, Obrenovich ME, Bragin V, Bragin I, et al. Oxidative stress-induced mitochondrial damage as a hallmark for drug development in the context of the neurodegeneration, cardiovascular, and cerebrovascular diseases. In Systems Biology of Free Radicals and Antioxidants. Springer-Verlag Berlin Heidelberg 2012. p. 2083-2126. 

5. Sasaki S, Iwata M. Mitochondrial Alterations in the Spinal Cord of Patients with Sporadic Amyotrophic Lateral Sclerosis. J Neuropathol Exp Neurol. 2007. 66(1):10-6. 

Lines 172-173: Features of autophagic vesicles (“double or multiple membranes”) are not convincingly demonstrated in Figure 2c, inset.

 - Replaced inset images with more recognizable or typical autophagic vesicles, and reworded the description to “… recognizable by their multivesicular myelin-like, or granular osmiophilic content…”, a description often used in literature 

1. Uchiyama Y, Shibata M., Koike M. et al. Autophagy–physiology and pathophysiology. Histochem Cell Biol. 2008. 129:407–420. 

2. Eskelinen EL, Kallio K. Ultrastructure of the Macroautophagy Pathway in Mammalian Cells. In: Loos B, Wong E (eds). Imaging and Quantifying Neuronal Autophagy. Neuromethods. Vol 171. Humana, New York, NY. 2022. p 13-21

3. Koike M, Nakanishi H, Saftig P, Ezaki J, Isahara K,Ohsawa Y, et al. Cathepsin D Deficiency Induces Lysosomal Storage with Ceroid Lipofuscin in Mouse CNS Neurons. The Journal of Neuroscience. 2000. 20:6898-6906. 

4. Eskelinen, EL. New Insights into the Mechanisms of Macroautophagy in Mammalian Cells. International Review of Cell and Molecular Biology. 2008. 266:207-247. 

Figure 192-3: Especially in Figure 3b and c, it is not clear why structures seemingly representing electron-lucent vacuoles are assigned as dilated Golgi apparatus.

 - The structures were labelled Golgi apparatus after consideration of several factors:

• Their perinuclear location.

• After thorough scrutiny of the cells, no continuity with the nuclear membrane could be found which might have indicated that the structures are smooth endoplasmic reticulum (ER).

• As the rough endoplasmic reticulum was largely unaffected, it is not expected that the smooth ER would show such a high degree of swelling, which convinced us that the “vacuoles” are not smooth ER.

• Even with the cisternae being dilated, in certain areas it can be seen that the “vacuoles” are stacked vesicular profiles.

• When compared to the Golgi apparatus of control cells, cells treated with lasalocid (which had less of an effect on the Golgi apparatus) or cells in which the Golgi apparatus structure could easily be discerned; the membrane is recognizable (electron density / thickness / folding) as those of the marked structures.

• Comparison to published images of dilated and swollen Golgi apparatus.

• And lastly; based on the knowledge of how ionophore exposure affect cells and what has been seen in other cell lines; swelling of the Golgi apparatus was expected and the finding in-line with that.

1. Radulescu A, & Siddhanta A, Shields D. A Role for Clathrin in Reassembly of the Golgi Apparatus. Molecular biology of the cell. 2007. 18:94-105. 

2. Berning L, Lenz T, Bergmann AK, Poschmann G, Brass HUC, Schlütermann D, et al. The Golgi stacking protein GRASP55 is targeted by the natural compound prodigiosin. Cell Commun Signal. 2023. 21: 275.

3. Podinovskaia M, Prescianotto-Baschong C, Buser DP, Spang A. A novel live-cell imaging assay reveals regulation of endosome maturation eLife. 2021. 10:e70982.

4. Zhang C, Rosenwald A, Willingham M, Skuntz , & Clark J, Kahn R. Expression of a dominant allele of human ARF1 inhibits membrane traffic in vivo. The Journal of cell biology. 1994. 124. 289-300.

Lines 214-215: It is not clear, what the bordered arrowheads are supposed to show in Figure 1e-1h. Line 156-157 indicate “filipodia-like and bleb-like surface structures” in Figure 1h that should be assigned.

 - Higher magnification images added as insets to Figure 1h to show the blebs and filipodia, and the arrowhead shape was changed to clarify which corner is the one pointing to the relevant feature (changed in all the images). Additionally, a description “..filipodia- and bleb-like surface structures..” was added to the figure legend.

Lines 227-228: Some of the arrowheads in Figure 3e clearly do not show surface indentations. In the results related to Figure 3, indentations are not covered at all as a feature of H9c2 exposed to salinomycin.

 - The pockmarked appearance is mentioned in lines 187-188 , we added clarification to the text to highlight that the pocked marked appearance is due to the observed surface indentations. The arrowheads were repositioned to make sure they are pointing to an example of a surface indentation and the arrowhead shape was changed to clarify which corner is the one pointing to the relevant feature.

Lines 253-254: At least Figure 6a seems not to show “myoblasts … filled with electron-lucent vacuoles…”.

 - Figure 6b replaced to show the cell filled with vacuoles, and moved to Figure 6a position, and the original Figure 6a was moved to the Figure 6b position. 

Lines 270-271: When comparing Figure 7a with Figure 6a and b, it seems not to be justified to state that “Myoblasts exposed to salinomycin were less vacuolated than those exposed to monensin (Fig 7 a)…”.

 - We exchanged figure 6a, which we hope better illustrate the difference between the two treatments. Additionally, the text was changed to clarify that less cells were extensively vacuolated, so it is not just the severity of the vacuolization it is also the number of cells affected.

Line 314: The structure encircled in Figure 8e is supposed to show apoptotic bodies, that seem to have a quiet different structure as compared with the apoptotic bodies shown in Figure 2g, 3g, and 6g. Please comment.

 - One difference between the images which creates the impression that the structure of these apoptotic bodies (ABs) are different is the magnification at which the ABs were imaged. Figures 2g, 3g, and 6g were imaged at roughly double the magnification of Figure 8e. When looking at another apoptotic L6 cell treated with lasalocid and imaged at a comparable magnification ()see below), the AB structure does look similar.

However, you are correct that there are some differences; 

1. Firstly; although the cell is undergoing apoptosis and covered in ABs, the spindle cell shape is still recognizable in Figure 8e and, 

2. Secondly when measuring the ABs in L6 cells treated with lasalocid, the ABs size mean is slightly smaller and the size range is also smaller, with a noticeable lack of larger ABs (>2 µm) 

H9c2 + monensin Size range: 1.6µm - 5µm Mean size: 2.3µm

H9c2 + salinomycin Size range: 0.7µm – 3.5µm Mean size: 1.1µm

L6 + monensin Size range: 1.0µm – 3.7µm Mean size: 1.5µm

L6 + lasalocid Size range: 0.6µm – 1.8µm Mean size: 1.0µm

The sizes of ABs in all of the exposed cells does still fall inside the general guide for ABs size (accepted to fall between 1µm - 5µm, however many cite 0.5µm - 5µm as a range and others have shown ABs as small as 100nm and as large as 10µm). 

The differences in size and size range observed between the different ionophores and cell types during this study can be contributed to the fact that the size and number of ABs formed is dependent on the cell type, the pattern, mechanism and the course of cell death (see references below). 

In conclusion we believe Figures 2g, 3g, 6g and 8e all represent ABs because: 

• The ABs / particle sizes are within range of known AB size

• TEM images show typical ABs along with other morphological apoptotic features. 

• The differences in ABs sizes was noticed and after review of the available literature, hypothesized to be due to the 

o different cell types which is seen to differ slightly in their response to different ionophores in this study.

o The treatment of that particular sample with lasalocid which is known, and also shown in this study, to have the least effect on the myoblast, which could contribute to a different pattern, or slower progression of cell death which would affect AB size. This would also explain why the apoptotic cell spindle shape is still recognizable.

This observed differences and the explanations thereof is outside the scope of this manuscript

1. Poon I, Parkes M, Lanzhou & Atkin-Smith G, Tixeira R, Gregory C, et al. (2019). Moving beyond size and phosphatidylserine exposure: evidence for a diversity of apoptotic cell-derived extracellular vesicles in vitro. Journal of Extracellular Vesicles. 2019. 8(1):1608786.

2. Crescitelli R, Lässer C, Szabó TG, Kittel A, Eldh M, Dianzani I, et al. Distinct RNA profiles in subpopulations of extracellular vesicles: apoptotic bodies, microvesicles and exosomes. Journal of Extracellular Vesicles. 2013. 2: 20677.

3. Xuebo X, Yueyang L, Zi-Chun H. Apoptosis and apoptotic body: disease message and therapeutic target potentials. Biosci Rep. 2019. 39(1):BSR20180992. 

4. Yu F, Yifan H, Di W, Bingdong S, Yan J, Xuefeng H, et al. Apoptotic vesicles: emerging concepts and research progress in physiology and therapy. 2023. Life Medicine. 2023. 2(2).

5. Núñez R, Sancho-Martínez S, Novoa J, et al. Apoptotic volume decrease as a geometric determinant for cell dismantling into apoptotic bodies. Cell Death Differ 2010. 17:1665–1671.

6. López-Hernández FJ. Cell Surface Area to Volume Relationship During Apoptosis and Apoptotic Body Formation. Cell Physiol Biochem. 2021. 55(S1):161-170

Line 316: Golgi apparatus seems hardly to be visible in Figure 8c.

 - We exchanged the image, and also added a higher magnification inset image to illustrate the Golgi apparatus.

Lines 351-354: An effect of monenesin on the Golgi apparatus is neither demonstrated in Figure 2 for H9c2 myoblasts nor in Figure 6 for L6 myoblasts.

 - The relevant passage has been reworded to highlight that the Golgi apparatus was not recognizable as stacked and organized cisternae after monensin exposure, and that vacuoles and vesicular profiles was all that was seen.

The discussion, while in general improved and much more interesting, is now a little bit lengthy and not always very stringent.

 - We rewrote the discussion section, focusing on the sections relating to published cell death mechanisms following ionophore exposure, condensing it and hopefully making it more coherent .

Minor points 

Line 151: should read: “in addition to” instead of “with some”.

 - Changed “with some” to “in addition to” 

Line 170, line 362: Please delete “with”.

 - Deleted “with 

Line 188-189: Please rephrase “…necrotic myoblasts were observed with little

to no membrane continuity, mostly only comprising cellular debris…”

 - Rephrased “…necrotic myoblasts were observed with little to no membrane continuity, mostly only comprising cellular debris…” to “ necrotic myoblast, primarily consisting of cellular debris and exhibiting a nearly complete loss of membrane continuity, were observed”

Lines 205 and 235: Please specify, what kind of surface structures are indicated by bordered arrowheads in Figures 4e-g.

 - Specified in the text and figure legend that the bordered arrowheads indicate filipodia- and bleb-like surface structures.

Line 214: Why are H9c2 myoblasts incubated in DMEM for 48 h termed as “negative control”?

 - In all the experiments performed the cells were cultured and grown in Dulbecco’s Modified Eagle’s medium (DMEM). In the exposed groups monensin, salinomycin, or lasalocid was added, whereas in the control group no ionophore was added and the cells were further incubated in the medium it was cultured in. As the control cells were not exposed to the treatment, and not expected to produce any results, they are negative-controls and serves as a baseline for the ultrastructure of “normal / healthy” cells.

Line 228: Should read “filipodia- and bleb-like surface structures” (line 199) instead of “surface structures”.

 - Filipodia- and bleb-like added to the figure legend.

Lines 220, 228, 300, and 314: Should 

---

## [Decision Letter · Decision Letter 2]

12 Sep 2024

Ultrastructural changes in cardiac and skeletal myoblasts following in vitro exposure to monensin, salinomycin, and lasalocid.

PONE-D-24-14459R2

Dear Dr. Lensink,

We’re pleased to inform you that your manuscript has been judged scientifically suitable for publication and will be formally accepted for publication once it meets all outstanding technical requirements.

Kind regards,

Hans-Peter Kubis, PD. Dr. rer. nat.

Academic Editor

PLOS ONE

Additional Editor Comments (optional):

Reviewers' comments:

Reviewer's Responses to Questions

**Comments to the Author**

1. If the authors have adequately addressed your comments raised in a previous round of review and you feel that this manuscript is now acceptable for publication, you may indicate that here to bypass the “Comments to the Author” section, enter your conflict of interest statement in the “Confidential to Editor” section, and submit your "Accept" recommendation.

Reviewer #1: All comments have been addressed

Reviewer #2: All comments have been addressed

2. Is the manuscript technically sound, and do the data support the conclusions?

Reviewer #1: Yes

Reviewer #2: Yes

3. Has the statistical analysis been performed appropriately and rigorously? 

Reviewer #1: N/A

Reviewer #2: N/A

4. Have the authors made all data underlying the findings in their manuscript fully available?

Reviewer #1: Yes

Reviewer #2: Yes

5. Is the manuscript presented in an intelligible fashion and written in standard English?

Reviewer #1: Yes

Reviewer #2: Yes

6. Review Comments to the Author

Reviewer #1: All comments have been addressed in the first revisied form of the manuscript, and this reviewer recommended the paper for publication.

Reviewer #2: All comments have been addressed. Therefore, the manuscript is now acceptable for publication in PlosOne

7. PLOS authors have the option to publish the peer review history of their article (what does this mean?). If published, this will include your full peer review and any attached files.

Reviewer #1: No

Reviewer #2: No

---

## [Editor Report · Acceptance letter]

16 Sep 2024

PONE-D-24-14459R2 

PLOS ONE

Dear Dr. Lensink, 

I'm pleased to inform you that your manuscript has been deemed suitable for publication in PLOS ONE. Congratulations! Your manuscript is now being handed over to our production team.

Kind regards, 

on behalf of

Dr. Hans-Peter Kubis 

Academic Editor

PLOS ONE